



# Climate engineering to mitigate the projected 21st-century terrestrial drying of the Americas: Carbon Capture vs. Sulfur Injection?

Yangyang Xu[1, *], Lei Lin[2, *], Simone Tilmes[3], Katherine Dagon[3], Lili Xia[4], Chenrui Diao[1], Wei Cheng[5], Douglas MacMartin[5], Zhili Wang[6], Isla Simpson[3], Lorna Burnell[7]

[1]Department of Atmospheric Sciences, Texas A&M University, College Station, Texas 77843, USA
[2]School of Atmospheric Sciences and Guangdong Province Key Laboratory for Climate Change and Natural Disaster Studies, Sun Yat-Sen University, Zhuhai, Guangdong, 519000, China
[3]Climate and Global Dynamics Laboratory, National Center for Atmospheric Research, Boulder, CO, USA
[4]Department of Environmental Sciences, Rutgers University, New Brunswick, New Jersey, USA
[5]Mechanical and Aerospace Engineering, Cornell University, Ithaca, NY, USA
[6]State Key Laboratory of Severe Weather and Key Laboratory of Atmospheric Chemistry of CMA, Chinese Academy of Meteorological Sciences, Beijing, China
[7]University of Nottingham, UK

*Correspondence to: Yangyang Xu (yangyang.xu@tamu.edu) or Lei Lin (linlei3@sysu.edu.cn)

**Abstract.** To mitigate the projected global warming in the 21st century, it is well recognized that society needs to cut CO2 emission and other short-lived warming agents aggressively. However, to stabilize the climate at a warming level closer to the present day, such as the "well below 2ºC" aspiration in the Paris agreement, a net-zero carbon emission by 2050 is still insufficient. The recent IPCC special report calls for a massive scheme to extract CO2 directly from the atmosphere, in addition to the decarbonization, to reach negative net emission at the mid-century mark. Another ambitious proposal is the solar radiation-based geoengineering schemes, including injecting sulfur gas into the stratosphere. Despite being in the public debate for years, these two leading geoengineering schemes have not been carefully examined under a consistent numerical modeling framework.

Here we present a comprehensive analysis of climate impacts of these two geoengineering approaches using two recently available large-ensemble (>10 members) model experiments conducted by a family of state-of-art Earth system models. The CO2-based mitigation simulation is designed to include both emissions cut and carbon capture. The solar radiation-based mitigation simulation is designed to inject the sulfur gas strategically at specified altitudes and latitudes and run a feedback control algorithm, to avoid common problems previously identified such as the over-cooling of the Tropics and large-scale precipitation shifts.



Our analysis focuses on the projected aridity conditions over the Americas in the 21st century, in detailed terms of the mitigation potential, the temporal evolution, the spatial distribution (within North and South America), the relative efficiency, and the physical mechanisms. We show that sulfur injection, in contrast to previous notions of leading to excessive terrestrial drying (in terms of precipitation reduction) while offsetting the global mean greenhouse gas (GHG) warming, will instead mitigate the projected drying tendency under RCP8.5. The surface energy balance change induced by Sulfur injection, in addition to the well-known response in temperature and precipitation, plays a crucial role in determining the overall terrestrial hydroclimate response. However, when normalized by the same amount of avoided global warming, in these simulations, sulfur injection is less effective in limiting the worsening trend of regional land aridity in the Americas, when compared with carbon capture. Temporally, the climate benefit of Sulfur injection will emerge more quickly, even when both schemes are hypothetically started in the same year of 2020. Spatially, both schemes are effective in curbing the drying trend over North America. However, for South America, the Sulfur Injection scheme is particularly more effective for the sub-Amazon region (South Brazil), while the Carbon Capture scheme is more effective for the Amazon region. We conclude that despite the apparent limitations (such as inability to address ocean acidification) and potential side effects (such as changes to the ozone layer), innovative means of Sulfur Injection should continue to be explored as a potential low-cost option in the climate solution toolbox, complementing other mitigation approaches such as emissions cut and carbon capture (Cao et al., 2017). Our results demonstrate the urgent need for multi-model comparison studies and detailed regional assessment in other parts of the world.

# 1 Introduction

The 21st-century global warming, mostly driven by $CO_2$ emissions and other greenhouse gas emissions, is one of the greatest crises facing the world today. A higher level of warming has been shown to lead to more frequent extreme weather and natural disasters (Schiermeier, 2011; Donat et al., 2013; Fischer and Knutti, 2015; Wang et al., 2017; Lin et al., 2018), all having profound implications for public health, agriculture and regional economy (Kunkel et al., 1999; Easterling et al., 2000; Knapp et al., 2008; Lesk et al., 2016). If left unchecked, temperature increases will soon pass the 1.5 and 2ºC warming levels (relative to pre-industrial era) in the coming decades and continue to rise (Peters et al., 2012), calling for a stronger need to comprehensively assess the ecological and societal impacts at various warming levels and plan for adaptation at regional to local scales.

To mitigate the future accelerated climate change, the importance of cutting $CO_2$ emissions and a few short-lived climate pollutants (SLCPs) has been well recognized (Meinshausen et al., 2009; Shindell et al., 2012; Victor et al., 2015). But most recent analyses show that even with a massive decarbonization to achieve a net-zero carbon emission by the mid-21st century,



it is still not sufficient to stabilize global warming at a level relatively close to the present level (e.g., the 1.5ºC goal as proposed in the Paris agreement) (Xu and Ramanathan, 2017; Miller et al., 2017). Instead, aggressive climate engineering schemes are needed (Keller et al., 2014; Tilmes et al., 2016; Lawrence et al., 2018). Among the many creative approaches (e.g. spraying seawater over the sea ice, ocean fertilization, marine cloud brightening, oceanic evaporation enhancement, and regional land albedo modification via vegetation or white roofs), two global-scale schemes have received the most considerable attention and even seed investment in carrying out proof-of-concept technical design (Vaughan and Lenton, 2011). The first is carbon capture and storage that directly extracts $CO_2$ from the atmosphere, especially close to its emission sources where the ambient concentration is high (Herzog, 2001). The second is the sulfur injection into the stratosphere to reflect sunlight to space (Crutzen, 2006). Both approaches, especially the first one, can be massively expensive, and at the same time, could induce side effects, including those not yet identified. More scrutiny on the effectiveness and undesired consequences is thus urgently needed before any real-world deployment.

The utility of sulfur injection as a means to mitigate climate change has been long suggested (e.g., Crutzen 2006, Robock et al., 2009; Niemeier and Tilmes, 2017), but some potential drawbacks have also been identified (Pongratz et al., 2012; Keller et al., 2014; Effiong and Neitzel, 2016; Irvine et al., 2017). For example, a notable downside of sulfur injection (mostly designed to be initiated from the Tropics to offset the higher intensity of solar radiation there), is that it tends to over-cool the Tropics, while under-cooling the polar warming due to the amplified greenhouse gas effect (Moore et al., 2014). To address these shortcomings, the strategic design has been proposed (MacMartin et al., 2017) to inject sulfur gas at multiple latitudes and selected heights to keep the north-south and Pole-to-Tropics temperature gradient largely fixed, while lowering the global mean temperature. By this design, global warming due to GHG will be balanced in a more spatially uniform fashion. Following the initial success of numerical tests in Kravitz et al., (2017), Tilmes et al. (2018, BAMS) conducted a large-ensemble model experiment (Geoengineering Largen Ensembles, GLENS) under the same rationale. A large-ensemble approach is particularly useful to assess climate impacts, because extreme weather can be better quantified with a larger sampling, and the role of natural variability internal to the climate system can be more robustly isolated.

Utilizing this GLENS dataset, recent studies have looked at hydroclimate change under this Geoengineering method. Based on the analysis from Simpson et al. (2019), the precipitation in tropical and extra-tropical regions shows a dry-get-wetter, wet-get-drier pattern due to the aerosol induced stratospheric heating. Cheng et al. (2019) also found the global land precipitation and evapotranspiration reduce slightly at the end of this century. Nevertheless, in both studies, the global soil moisture is well maintained by the geoengineering method except for India and the Amazon. Cheng et al. (2019) explained that the reduction of summer soil moisture in India and the Amazon is dominated by the decrease in precipitation.

Our study extends the scope of existing GLENS analysis, by placing the Sulfur Injection into the greater context of climate

engineering (MacMartin et al., 2018, PTRS). Here we aim to compare the multiple metric of mitigation "benefits" of Sulfur Injection with Carbon Capture, another more expensive and yet more fundamental approach (e.g., to mitigate ocean acidification). Because many other analyses using GLENS have documented the global impact of Sulfur Injection, here we choose to focus on a smaller region of North and South America, and on a specific set of land-based hydroclimate quantities that are of close relevance to agriculture, ecosystems, socioeconomics, and indirectly related to the terrestrial carbon cycle in

the Amazon and its feedback to climate change.

The structure of the paper is the following. In Section 2, we provide the details of the model description, experiment setup, the observational dataset used as model calibration, and the rationale of deriving climate benefits. In Section 3, we present and compare the climate benefits of two geoengineering approaches on a global scale. The regional changes are presented in

Section 4 with detailed discussions on the mechanisms of the distinct results over the sub-Amazon and Amazon regions. In Section 5, we discuss the results of climate responses in terms of normalized changes with respect to global temperature change. Section 6 further discusses the implications and makes concluding remarks.

## 2 Methods

### 2.1 Global Climate Models

Two fully coupled ocean and atmospheric models, which are identical for the ocean, and tropospheric atmosphere component, are used in this study. The horizontal resolution of the atmosphere component for both models is set at $0.9° \times 1.25°$.

The major difference between the two models is the stratospheric atmosphere component, and the land components are also

slightly different, which are discussed below.

The CESM1-WACCM (the Community Earth System Model version 1 with the Whole Atmosphere Community Climate Model, short as "WACCM" hereinafter) has a high-top configuration that extends to about 140km ($6x10^{-6}$ hPa) and has a comprehensive representation of stratospheric chemical and dynamic processes (Mills et al., 2017), both crucial to simulate

the radiative impact of stratospheric aerosols (via volcanic or artificial injection). The sulfur injection simulation is only available using this model version.

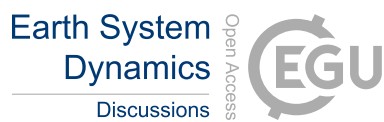

CESM1-CAM5 (the Community Earth System Model version 1 with Community Atmosphere Model version 5, short as "CESM" after this) has a typical model top at around 40 km. The CO2 mitigation simulation is only available using this model

version.

## 2.2 Numerical Experiments

a. Stratospheric Sulfur Injection

The simulations were conducted with WACCM: one with baseline emissions (RCP8.5) and the other with Sulfur injection applied at 2020 and afterward (hereafter referred to as Sulfur Injection). The amount and location of Sulfur Injection were

strategically selected to stabilize the climate at the present-day level and minimize the undesired perturbation to the latitudinal temperature gradient (Tilmes et al., 2018). The simulation was repeated multiple times (see Table 1 for details) to form a larger ensemble, which enhances the statistical significance of the assessed climate impact, especially at a regional level.

b. Carbon Capture and Storage

Similarly, two simulations were conducted with CESM: one with baseline emissions (RCP8.5) and the other with $CO_2$ capture and massive $CO_2$ emissions cuts applied to start in 2015-2020 (hereafter referred to as Carbon Capture). Net-Zero carbon emission is thus reached at 2050, beyond which the net emission turns negative, which helps the planet to stabilize at 1.5ºC warming, just slightly higher than the present-day (2010-2019) temperature level. We show the mass amount of CO2 captured (although in CESM simulation, the corresponding CO2 concentration is prescribed rather than simulated) and the associated

negative radiative forcing in later figures to illustrate the impact of these idealized trajectories. However, the technical feasibility and required socioeconomic shift (Sanderson et al., 2017) is beyond the discussion of this paper.

## 2.3 Deriving the climate responses

With the two sets of paired simulations, the effect of the two geoengineering approaches can be assessed by contrasting the run with geoengineering applied with the baseline potential warming worlds, which we would refer to as the climate "benefits."

Of course, for specific regions and certain climate variables, it may turn out that the climate "benefits" are negligible or even negative – i.e., geoengineering makes the climate change under RCP8.5 even worse.

## 2.4 Hydroclimate variables examined

In this study, we focus on climate quantities over land due to their close relevance to agriculture, ecosystems, and the carbon cycle, which include:

50        a.    T: surface air temperature at the reference height of 2 m above the ground.



b. P: total precipitation, which includes rainfall and snowfall without differentiating large-scale or convective component as formulated in the model.

c. PET: potential evapotranspiration, which represents the atmospheric demand for water. PET is calculated using model output, following the physically-based Penman-Monteith equation (Shuttleworth, 1993), which includes key

parameters of surface air temperature, surface available energy, surface wind speed, and relative humidity.

d. Aridity                         Index                           (P/PET).
The aridity index is calculated as the simple ratio of annual P and annual PET (Mortimore et al., 2009; Hulme, 1996; Middleton and Thomas, 1992). P/PET is highly correlated with vegetation types (Feng and Fu, 2013; Huang et al., 2017). Using the P/PET classification by the United Nations Convention to Combat Desertification [United Nations

Convention to Combat Desertification, 1994], drylands (characterized by P/PET <0.65) further classify various types, including sub-humid (0.5< P/PET <0.65), semi-arid (0.2< P/PET <0.5), arid (0.05< P/PET <0.2), hyper-arid (P/PET <0.05).

e. Other variables to depict aridity and corroborate Aridity Index.

To further test the utility of P/PET as a metric to depict aridity conditions, we show the similar temporal evolution of P/PET with soil moisture, and that between PET and actual ET (another metric commonly used) over major land regions (60ºS to 60ºN) in Figure 1. Spatial patterns also show similar agreement between P/PET and soil moisture as well as P-ET (another metric commonly used) for both present-day and future change (Figure 2), again supporting the appropriateness of using P/PET as an aridity metric.

## 70   2.5 Bias correction of the raw model results

Climate model output cannot be taken at its face value. The analysis, especially regarding the future projection, must consider the potential model bias in simulating the past and the present as well as the uncertainty inherent to climate models that are so far poorly constrained. Considering these, we corrected the model simulated T, P, and PET based on the observational dataset, following the approaches in Dai (2011) and Feng and Fu (2013). The NOAA Climate Prediction Center (CPC) data (Fan and

van den Dool, 2008; Chen et al., 2002) is used as the benchmarking temperature and precipitation over land. The Global Land Data Assimilation System (GLDAS) (Todell et al., 2004) provides the surface sensible heat flux and latent sensible heat flux, specific humidity, and wind speed to derive the observation-based PET as in Feng and Fu (2013). The climate variables over the ocean are not corrected and thus not analyzed in this study.

In a nutshell, the temperature is adjusted using the addition/subtraction of identified biases on a monthly basis: T_adjusted = T_raw − (T_model − T_obs). For P or PET, the variables are adjusted using multiplication/division of identified biases at a





monthly basis: P_adjusted = P_raw / (P_model /P_obs). A multiplication/division is used instead of addition/subtraction to avoid the negative numbers for P or PET.

The bias correction based on the observational dataset is only directly applied to CESM_Historical (with 30 runs from 1975 to 2005). The bias correction will force the model output to agree with the observed 1975-2005 climatology (and removing the bias of the seasonal cycle if any), but not necessarily agree with the observation at every year.

In order to make sure the magnitude of changes are consistent between different set of simulations (CESM vs. WACCM), the

same bias correction is applied to future simulations (CESM_RCP8.5 and CESM_CarbonCapture, in 2006 and afterward), which branched from the CESM_Historical at 2005, without adjusting the simulated year-to-year variability in coming decades. For WACCM_RCP8.5 simulations, we indirectly adjust them to agree with the corrected CESM_RCP8.5 in 2010-2019 (the "present-day" as defined in this study). We could not adjust WACCM directly based on observations because WACCM-RCP8.5 (20 members) simulation only starts in 2010 and overlaps with the entire observation for less than ten years. A single

run of WACCM during 1980-2010 (not used in this study) is available but does not provide the sufficient ensemble size as in CESM_Historical (1975-2005). Similarly, the WACCM_SulfurInjection was adjusted to match with WACCM_RCP8.5 in 2020 (when the WACCM_SulfurInjection simulation branched from WACCM_RCP8.5 due to the deployment of sulfur injection). The indirect approach is mathematically equivalent to the direct correction towards the observational dataset if the same base period was used (verified later in the text).

.00

## 3 Mitigation at the global scale: P vs. P/PET and the response time scale

### 3.1 Temperature

In Figure 3, we show the long-term changes simulated by all four model experiments (two baseline worlds and two mitigated climates via different geoengineering schemes). We focus on the following three periods: present-day (2010-2019), mid-

.05    century (2046-2055), and end-of-century (2086-2095). Without the bias correction, the models do not agree on climatological T and P for the recent period of 2005-2020 (Figure 3, a-b) and the future changes (also seen in the regional map in Figure 2, the first row). Therefore, it is essential to apply the bias correction to model output before carrying out a meaningful comparison of future changes.

.10    Comparing the bias-corrected model output (Figure 3, c and d), because of the larger climate sensitivity, WACCM warm up faster than CESM, reaching a 6ºC warming under RCP8.5 just within this century (T (2086-2095) - T(2010-2019), which is



~7ºC relative to pre-industrial, thus posing an existential threat to mankind if it comes true (Xu and Ramanathan, 2017). The CESM with a climate sensitivity of 4ºC shows a 4.7ºC warming during the 21st century. The two types of mitigation efforts, by design, would lead to a similar amount of temperature stabilization to a level close to the present-day.

The Sulfur Injection simulation here leads to a cooling of 6ºC towards the end of the century, compared with the baseline warming, larger than many previous Sulfur Injection model experiments. This larger cooling is partially due to the large injection amount, but also due to the design that the injection location is off the tropics, producing a greater temperature response than the injection solely from the tropics (such as those simulated by the same model in Tilmes et al., (2017) and Kravitz et al., (2018).

The mid-century warming over global major land projected by those two models amounts to 2.0 to 2.5ºC compared to the present-day (Table 2). Sulfur Injection can completely offset the warming by 2.4ºC, while Carbon Capture can only mask about 70% of 2.0ºC by 1.3ºC. Even though both schemes are designed to be deployed in 2020, the longer lifetime of $CO_2$ makes the impact of perturbation to the carbon cycle and atmospheric concentration slower to emerge. This additional inertia (or lag) in $CO_2$ emission-based mitigation is well noted when the $CO_2$ emission cut was previously contrasted with the mitigation of short-lived climate pollutants (e.g., Hu et al., 2013). Towards the end of the century, the lagging effects in the Carbon Capture case become negligible, and both schemes can almost completely offset the projected warming in the baseline by 4.0ºC and 5.7ºC, out of the 4.5ºC and 5.8ºC baseline warming, respectively (Table 2, column 1 right). Also note that the fractional reduction due to two types of geoengineering schemes is larger if only the land variables are considered (Table 2, Column 3).

## 3.2 Precipitation

Similar lagging effects are also found for global land precipitation. Table 2 shows that the mid-century reduction due to Carbon Capture is 31% of the projected increase (23.0 mm/year), and the Sulfur Injection can reduce the precipitation increase by as large as 75%. Again, towards the end of the century, the difference between the two cases, in terms of mitigation potential, becomes smaller (62% reduction by Carbon Capture and 98% by Sulfur Injection).

The close to 100% reduction of precipitation due to Sulfur Injection is worth commenting on. Because of the lack of direct radiative heating in the troposphere, such as due to $CO_2$ or black carbon aerosols, sulfate aerosols will have a larger precipitation effect (per degree of global mean temperature change) compared to $CO_2$ (Lin et al., 2016; 2018). Another alternative perspective to explain the over-drying (or slow-down of the hydrological cycle) due to geoengineering is the increase in atmospheric stability and suppressed evapotranspiration, especially in the tropics which tends to be over-cooled in

previous Sulfur Injection model experiments. As a result, when Sulfur Injection is carefully introduced to balance the CO2 warming, the total rainfall will usually be excessively suppressed (so-called "over-drying") (e.g., Jone et al., 2013; Tilmes et al., 2013; Crook et al., 2015).

However, in this WACCM setup, the over-drying consequence, especially over the Tropics, is dampened. This is likely due to the following reason. Because of the careful experiment design here, Sulfur Injection, which is more greatly introduced to the extratropical region, induces a much reduced tropical overcooling compared to previous stratospheric geoengineering experiments. Indeed, the global mean temperature, as well as the north-south gradient, will be well preserved while offsetting the baseline CO2 warming.

The relatively weaker tropical cooling (with a larger forcing off-tropics) leads to a smaller over-drying, despite a persistent suppression of large convective precipitation (see Simpson et al. 2019). Indeed, the Sulfur Injection mitigated climate will still see a slight reduction in precipitation by 10-20 mm/year (WACCM_SulfurInjection (2086-2095)- WACCM_SulfurInjection (2010-2019), Figure 3d including the ensemble spread), over the majority of the land. The over-drying is stronger (30 mm/year) if the model output is not bias-adjusted (Figure 3b). This over-drying, despite small in this model, is considered as a side effect of Sulfur Injection and often used as an argument against its deployment. In contrast, the Carbon Capture falls short in offsetting the full magnitude of projected precipitation increases (green line in Figure 3b, d), and thus will not run into the problem of "over-drying".

The applicability of this precipitation-centered argument will be put into the test in the next subsection.

## 3.3 PET and P/PET

It is increasingly well-recognized that precipitation alone does not reflect the full effects of the hydrological cycle on the terrestrial ecosystem. A full suite of aridity indices has been developed and is continuously being improved to provide a better depiction of aridity conditions (Mishra and Singh, 2010). It is beyond the scope of this paper, which emphasizes the contrast of two mitigation schemes, to discuss the advantages and limitations of those approaches. However, we aspire to include more than one indicator from the model output to examine the robustness of our results largely based on P/PET. We showed that ET and soil moisture are closely correlated with PET and P/PET, respectively, over the global scale (Figure 1). Moreover, Figure 2 shows that a close agreement between P/PET, soil moisture, and P-ET exists not only for the present-day climatology but also for the future increase under the warming scenarios.



The major difference between the two geoengineering approaches, when viewed on a global scale, is related to PET. The projected PET growth approximately scaled with T increase, is the main reason why the aridity will be exacerbated in the future despite an increase in P (Fu and Feng, 2014, JGR; Scheff and Frierson, 2014). Indeed, the projected P/PET (Figure 3f) is decreasing over the major land in both sets of model simulations, with a larger magnitude in CESM than WACCM (also seen in the last two columns of Table 2). The two geoengineering schemes can both limit the PET growth (Figure 3e) to bring forth the benefit of curbing aridity worsening, but with considerably large differences in the magnitude and timing, as detailed below.

The Carbon Capture can lower the projected increase in PET by 60% to 83% at the mid-century and end of the century, respectively, which leads to an almost complete offset of P/PET decline (see green line of Figure 3f). Therefore, the mid-century and end-of-century P/PET are mostly the same as the present-day value, except for the near-term (prior to 2040) drop in P/PET due to the lagging effects of $CO_2$ mitigation which will be elaborated later (Figure 4).

On the other hand, Sulfur Injection leads to a major decrease in PET that more than offsets the projected increase due to GHG warming (Figure 3e). For example, the mid-century projected PET increase is 102.2 mm/year, but the Sulfur Injection can lower that by 127.8 mm/year, which drops the absolute value of PET by 25.6 mm/year. The larger reduction in PET due to Sulfur Injection also reflects in the larger increase in P/PET by 0.03 to 0.05 (presumably shifting the climatic zone to a category beneficial to the cropland), despite a smaller projected P/PET decline in the baseline WACCM warming.

The large response of P/PET due to Sulfur Injection to more than offset the baseline changes can be clearly seen in Figure 3f (blue line). Different from all other curves, Sulfur Injection mitigated P/PET features an increase in absolute value, making the geoengineered climate over land overall "wetter" than the present-day, despite a reduction in precipitation. Although both types of geoengineering can stabilize climate at a level close to present-day, Sulfur Injection appears to have this additional "benefit" of reversing the drying trend projected in the baseline warming scenarios.

The increase in P/PET is mostly achieved in the near term (before 2050), due to the shorter response time associated with Sulfur Injection. Figure 4 shows the "benefit" or mitigation effects (the difference between two simulations) in percentage values relative to present-day climatology. This highlights the response time of various approaches in the near-term and long-term. During the first 30 years after the deployment, Sulfur Injection will lead to a quick reduction of PET, and thus actually cause the P/PET to increase. After the first few decades, when the surface cooling starts to emerge, the precipitation also weakens in response to Sulfur Injection. As a result, the P/PET increase in response to Sulfur Injection will slow down after 2050 (with the rate dropping by more than half from 1.5 %/decade to 0.7 %/decades). This non-monotonic behavior in Sulfur



05 Injection induced responses to P/PET are distinguished from the Carbon Capture case, which in contrast, always falls behind the Sulfur Injection changes in terms of T, P, and PET. In the case of P/PET, the increase due to Carbon Capture (when compared relative to the baseline warming, not in absolute values relative to present-day), only start to be significant toward the end of the century.

## 3.4 Summary

10 In terms of mitigation at a global scale, we emphasize P/PET is closely tied to land aridity conditions (Figures 1 and 2) and vegetation response (Huang et al., 2017), rather than P itself. Our results here suggest that Sulfur Injection not only offsets the projected worsening in aridity in the 21st century, but also, surprisingly, leads to recovery from the present-day condition, which is already worse than pre-industrial conditions or even just decades ago.

15 More encouraging, the recovery ("wetting" over major global land) is detectable in the near term (MacMartin et al. 2019). Carbon Capture, in contrast, has a weaker (even in relative terms) and slower mitigation potential and does not lead to the reversal of P/PET as in response to Sulfur Injection. The response time of these two approaches is illustrated in Figure 4e and f. While the CO2 captured has a quick increase after the deployment (2020), the radiative forcing due to the reduced CO2 concentration takes a slower trajectory to increase and is always falling behind the negative forcing due to Sulfur injection.

The quicker and stronger response to Sulfur Injection, suggests a benefit from the deployment, providing a further incentive for adopting this scheme. However, because of the quicker response, one could counter-argue that Sulfur Injection should be reserved as the "last resort" option only used when the CO2 warming becomes too large towards the end of the century.

## 4 Regional change: focusing on the land of the Americas

How do the results and arguments above hold over specific regions? Next, we present a detailed comparison over North America, and South America and discuss the mechanism. Other recent works have studied the precipitation response in various regions. Cheng et al., (2019), using soil moisture as the metric, have found major regions that would benefit from the Sulfur Injection in having the aridity trend reverted. However, certain regions will have the aridity conditions worsened in the Sulfur 30 Injection based geoengineered scenario. Here we focus on North and South America using both P and P/PET to depict the full picture of hydrological cycle response.

### 4.1 North America

The main reason for focusing on North America is the arid land expansion from American West into the Great Plains. The so-called 100th meridian, which traditionally separates the dry and mild climate, will likely shift eastward in future climate (Seager et al., 2018). This is clearly shown in the projection of both models with a major decrease in P/PET over the Great Plain regions by 10% to 20% (Figure 5 d,g). The corresponding change in absolute terms of P/PET is shown in Figure 2. The slight increase (light blue) over the intermountain western US regions does not pass the significance test in being different from the baseline simulations.

For North America, it is clear that the Sulfur Injection has a stronger effect than Carbon Capture, especially for the Central US in the mid-century (Figure 5 f vs. e). Towards the end of the century, the Sulfur Injection increases P/PET by 0.05 (7.2%), while the Carbon Capture can eliminate the projected decrease in P/PET (Figures 5 and 6). This is largely consistent with the results of Eastern US (although to a lesser extent) and the global value (Figure 6, also illustrated in the time series of Figure 3).

The large decrease in PET in response to Sulfur Injection is the main reason for a slight increase in P/PET compared to present-day conditions in the geoengineered cases. The P/PET changes over the US are consistent with the tendency of the global major land (Figure 6d), and also largely supported by other metrics (soil water and P-ET in Figure 6 e,f).

Another way to examine the severity of aridity evolution is by further classifying the dryland into a few sub-types. These subtypes are classified by P/PET values, which are shown in the color bars in Figure 5 first row). The arid area (P/PET ranges of 0.2 to 0.5) would increase from 15 million km2 in the historical period to 16.6 million km2 under the warming at the end of the century. The Carbon Capture can revert it to 15.1 million km2, but Sulfur Injection can more than offset it to lower it to 13.5 million km2. Over North America, the results are generally similar, with a stronger reversal of semi-arid area (P/PET ranges of 0.5 to 0.65) due to Sulfur Injection than Carbon Capture.

## 4.2 South America

The main reason for focusing on South America is the drying over the Amazon (by more than 30% using P/PET as the metric) could lead to die out of the rainforest and decrease the carbon sink and cause an increase in forest fires– a major positive feedback that's missing in most of the global model projections so far, including the ones presented here.

Unlike the consistent tendency over North America, largely representative of global land change, South America's responses are more complicated. Notably, there is a major difference between northern Brazil, where the Amazon rainforest is mostly located, and the southern part of the nation (green and purple boxes in Figure 5a).


Northern Brazil will see a major decline of P/PET of 0.2 to 0.4 (15 to 30%) in the baseline warming cases (Figure 5g), the largest among the four regions we considered in this study, transitioning the wetland into semi-arid dryland. This large decline of P/PET is a result of both PET increase as in other regions, but also P decrease unique in this tropical land, in contrast to other regions examined. If the RCP8.5 projection turns out to be true for the 21$^{st}$ century, a major consequence is the fate of the rainforest and its carbon uptake capacity. Given the high stakes, a relevant question is whether the two geoengineering
schemes are able to revert the drying trend over the Amazon.

The CESM results show that Carbon Capture can essentially overturn the baseline drying and keep the P/PET at the same level of present-day (Figure 5). This is achieved by both offsetting the precipitation decline as well as the PET increase (Figure 6 b,d). The Carbon Capture (green in Figure 6b) mitigated climate will lead to a small increase in precipitation (relative to
present-day) in northern Brazil, associated with a much smaller regional warming (<0.5ºC) than baseline (4.5ºC) (Figure 5a). Note that the regional precipitation change can be hard to project at the sub-continental scale, and thus, the CESM simulated Carbon Capture response could be model dependent.

On the other hand, the WACCM results show that Sulfur Injection falls short of completely overturning the drying tendency
over Northern Brazil in the baseline warming but can mitigate the drying magnitude by 60% by the end of the century. The reason for the insufficiency of Sulfur Injection is that, despite a substantial reduction in PET (Figure 6c, going from 0.8 to 0), the WACCM results continue to produce a stronger precipitation reduction (relative to present-day) in the case of Sulfur Injection (blue dots in Figure 6b), unable to revert the precipitation reduction in the warming baseline. This additional suppression of tropical rainfall due to Sulfur Injection is consistent with results shown in Simpson et al., (2019), which
indicated that the dynamical response to stratospheric heating played an important role in this during JJA (but not in DJF).

Although the simultaneous reduction of PET in response to Sulfur Injection leads to the end results of "wetting" (in terms of P/PET increase (brown and blue points in Figure 6d), the magnitude (a roughly 50% offset) is small compared that due to Carbon Capture (an offset by almost 100%, green and red dots in Figure 6d), in which the precipitation is reversed to increase
slightly, more than just reverting the precipitation decrease in the warming baseline (green in North Brazil, Figure 6b). Overall there is weaker mitigation of the P/PET drying trend in response to Sulfur Injection compared to Carbon Capture over Northern Brazil. However, interestingly, the opposite is true for the sub-Amazon region, detailed below.

The Carbon Capture continues to be capable of mitigating the projected drying trend almost entirely (similar to global and
eastern US case). Despite being smaller than the Sulfur injection, there is sufficient capacity of Carbon Capture in overturning





the drying trend over the sub-Amazon (from a 0.08 reduction of P/PET to almost zero), again due to a large reduction in PET, despite slightly weakened precipitation. This suggests again that focusing on P alone may lead to a biased interpretation of the projected and mitigated trend.

Notably, Sulfur Injection induces a much larger increase in P/PET ("wetting"), compared with the effect of Carbon Capture, actually offsetting the projected P/PET decrease by 150%. This is due to a regional increase in precipitation, which adds on top of PET reduction, to cause a larger increase of P/PET. The mechanism is exactly the flip side of Sulfur Injection's smaller capacity (by only 60%) to revert P/PET decline over the Amazon (Northern Brazil), where the precipitation is projected to further decrease in response to Sulfur Injection, despite a strong local cooling effect.

Synthesizing the precipitation changes over the Amazon (Northern Brazil) and sub-Amazon (Southern Brazil), the precipitation (especially JJA season as in Figure 5 of Cheng et al., 2019) is shifted from the deep tropics to the subtropics in response to Sulfur Injection, leading to an actual increase in precipitation over the sub-Amazon by 36.5 mm/year, rather than just reversing the 18.2 mm/year decrease in the baseline warming case.

A substantial reduction in PET over sub-Amazon (but consistently true for other sub-regions explored in Figure 6), working together with the precipitation increase, would induce a large increase in P/PET (0.2) relative to the baseline case, more than enough to offset the projected decrease of 0.15 and actually to lead to P/PET increase by 0.05. Considering that this region at the present-day is already subject to a detectable drying, such an over-compensation in P/PET in response to Sulfur Injection
should be viewed as beneficial to the regional climate and ecosystem, a point we have stressed for the global mitigation results, but want to echo again here for sub-Amazon regions.

## 4.3 Summary

Overall, over three out of four regions examined here (Western US, Eastern US, and Southern Brazil, except for Northern Brazil), Sulfur Injection has a larger capacity in not only mitigating but also flipping the projected drying tendency. This is
evidenced by the large response seen in the spatial maps of Figure 5 (especially the larger response in the mid-century due to time lag effects discussed in Section 3.1). A complete reversal or even improvement of P/PET drying highlights the benefit of Sulfur Injection, at least for these three regions. On the other hand, Carbon Capture can almost completely offset the P/PET change by 100%, also the desired outcome. The mechanisms to achieve this complete offset can vary from one region to another, sensitive to the regional precipitation projection in the baseline and mitigated scenarios (e.g., over the western US),
and thus potentially sensitive to model configurations. The responses over these three regions are found to be generally

representative of the global mean (e.g., Figure 6 "major land" results with smaller error bars), which are presumably less subject to model diversity issues.

The South America regions present a clear argument that a precipitation-centered viewpoint may mistakenly overestimate the Sulfur Injection side effect in causing aridity by ignoring its accompanying suppression on PET. The contrasting results of Sulfur Injection to overcome the global drying tendency, when using precipitation only or using P/PET, is clearly demonstrated for the Amazon region, but also similarly true for other regions such as the Eastern US (to a lesser extent). For example, in Figure 5, the precipitation change over the Eastern US is slightly negative compared to present-day ("over-drying") but become slightly positive in P/PET, suggesting an actual "wetting", which is an improvement from the drying condition already detectable at the present-day (2010-2019) compared to pre-industrial era.

One exception to the summary above is Northern Brazil (Amazon region). The Sulfur Injection, but not Carbon Capture, can only partially offset the projected drying, due to a further decrease in precipitation locally in the deep tropics (featuring a shift from the deep topics to subtropics), and a smaller amount of PET reduction from the baseline increase (Figure 6c), compared to the other three regions and global mean, presumably due to extensive cloud cover over the tropical land but could also be due to the vegetation cover difference between Northern Brazil and Southern Brazil.

## 5 Normalized changes with respect to global temperature change

The absolute value of climate benefits (including those presented in Table 2 and Figure 6) is dependent on the strength of geoengineering measures and thus can be less useful in a broader context. What is more informative is the physical mechanism governing the changes. In order to elucidate the different mechanisms contributing to the simulated changes over North and South America, we next examine the individual climate variable's contribution to P/PET changes, in a normalized (%/ºC) perspective.

In the following analysis, we show that even though Sulfur Injection has a larger mitigation capacity as detailed in the previous sections, it is mainly due to the larger forcing introduced and the larger global cooling realized (close to 6ºC). Sulfur Injection is less effective in mitigating the aridity worsening when the mitigation is normalized by the global cooling realized.

Figure 7 shows that the normalized changes have largely the same spatial pattern as in the absolute values presented in Figure 5, but the Sulfur Injection induced changes are now of similar magnitude compared to the Carbon Capture case for both time periods, with a slightly larger value for the mid-century but smaller values at the end of century. Note that because of the





normalization, some of the Amazon and sub-Amazon regions have changes less than 1%/°C (white area in Figure 7), in response to Carbon Capture and Sulfur Injection, suggesting an insignificant change of aridity condition if the global mean temperature mitigation is smaller than 1°C. The Great Plain of the US, however, would benefit from either of the two
·60    geoengineering schemes, with a change as large as 5%/°C to 10%/°C.

What explains the weaker normalized P/PET response of Sulfur Injection? The relationship of various quantities (P, PET, P/PET) with respect to global mean temperature changes is further illustrated in the scatter plots (Figure 8), with the slope representing the sensitivity of P, PET, P/PET to global cooling induced by the two geoengineering schemes (4 to 6°C of cooling
·65    in the x-axis). The calculated slope using the "linear fit" in Figure 8 is summarized in Table 4 and compared with another method of deriving the sensitivity, using the epoch difference between the two end-of-century contrasting "epoch." Note we have also tested the robustness of the linear regression approach by using decadal mean (as in Table 4) or annual mean (as in Figure 8), which turns out to yield a small difference. For example, the precipitation sensitivity changes from 2.6 %/°C (Figure 8a) to 2.7 %/°C at a global scale (Table 4).

·70

The larger sensitivity in precipitation due to sulfur injection is well understood and consistent with our previous results of tropospheric SO2 and CO2 increase. The larger slope of precipitation for Sulfur Injection is expected, but the magnitude is smaller than those in our previous studies (Lin et al., 2016) using a different model (CESM1) and considering the 20th-century tropospheric aerosols changes (Table 4). In contrast, the Carbon Capture induced precipitation response (1.1 %/°C) is of similar
·75    magnitude when compared with the 20th century CO2 increase examined with the same model (CESM1; Lin et al., 2016). Thus, we conclude that the model dependence of precipitation response is likely the largest uncertainty of the present analysis and should be revisited with a multi-model ensemble.

Different from the precipitation sensitivity, the PET sensitivity to Carbon Capture and Sulfur Injection are similar (3.8 %/°C
·80    vs. 4 %/°C). A similar slope for PET, when combined with the greater amount of precipitation decrease, will lead to a smaller increase of P/PET in response to Sulfur Injection given the same amount of cooling. In other words, the almost identical PET sensitivity in response to Sulfur Injection and Carbon Capture is the main reason the Sulfur Injection has a smaller influence on P/PET (by a factor of two when using linear regression as in Figure 8, -1.1 %/°C vs. -2.2 %/°C). If the Sulfur Injection leads to a greater amount of surface radiation reduction in other models such as CESM1 as in Lin et al., (2016), the P/PET sensitivity
·85    due to Sulfur Injection could be as large as the Carbon Capture case here. For example, we previously found that the tropospheric SO$_2$ will lead to a much stronger PET sensitivity (6.3 %/°C) than CO$_2$ (4.6 %/°C) because of the surface solar radiation reduction. In this study, the PET response to Sulfur Injection (4 %/°C) is only slightly higher than that due to Carbon Capture (3.5-3.8 %/°C depending on the method of normalization).



90 Another complexity arises. Our previous 20th century based analysis shows that tropospheric sulfate-induced PET change strongly by 6.3 %/°C. However, it still falls short of compensating the large precipitation decrease by 6.7 %/°C, thus producing a small decline of P/PET (by only 0.4 %/°C) in response to tropospheric sulfate cooling, opposite to the larger increase of P/PET in response to stratospheric Sulfur Injection (-1.5 %/°C). The current model experiment precludes a definite answer to understanding the discrepancy of Sulfur induced P, PET, and P/PET responses, because of model difference (WACCM here

95 vs. CESM in Lin et al., 2016) and forcing difference (stratospheric here vs. tropospheric in Lin et al., 2016). We speculate that the difference is mainly from the forcing difference because the stratospheric aerosol may be less effective in changing surface energy balance than tropospheric aerosols, which can induce warm cloud changes.

The speculation above is worth testing, for example, using the volcanic forcing experiment in the Last Millennium Ensemble

(LME, using CESM but with 2° resolution; Fu et al., 2017). Our preliminary analysis shown in Table 4 supports this argument. The GHG-only and Aerosol-only results from LME are similar to Lin et al., 2016's results of CO2- and SO4-induced responses using the same model (but with 1° resolution). The volcanic eruptions (active during the three periods of 1250-1270/ 1450-1460/ 1800-1820) before 1850 induced much weaker P response (2.5 to 4 %/°C) than tropospheric SO2, similar to the Sulfur Injection results obtained here (but using the different model of WACCM). Also similar to the Sulfur Injection results here, despite the model difference, is the overall increase in P/PET when stratospheric aerosol cooling is imposed (i.e., negative

sensitivity of P/PET in Table 4), suggesting the "benefit" of reversing the projected P/PET decrease via stratospheric Sulfur Injection could be model-independent.

In Table 5, we further break down the induced PET and P/PET changes (%/°C) into four key quantities that contribute to PET

calculation. The surface wind and relative humidity are, in general, smaller contributors compared with T and surface energy, except that at the regional level. RH can increase in response to cooling and thus reduces the PET. The stronger surface energy perturbation due to Sulfur Injection compared to Carbon Capture, mainly due to the strong suppression of incoming solar radiation reaching the ground, is clearly a major contributor to the larger reduction of PET at a global level, but not so for South America, presumably due to the extensive background cloud covers. Similarly, the P changes over South America are

complicated, featuring a decrease of North Brazil rainfall given the Sulfur Injection cooling (0.9 %/°C) but an increase in response to the Carbon Capture cooling.

To summarize this section, the weaker sensitivity of Sulfur Injection, in addition to its weaker absolute capability over the Amazon as examined in detail before, are the two major counterarguments against the effectiveness of Sulfur Injection.

However, the inter-comparison with our earlier studies on tropospheric SO2 increase and Volcanic response using different



configurations (20th century or Last Millennium; CESM1 with 1º or 2º resolution) suggest the qualitative robustness of the results and common physical mechanisms.

## 6 Concluding Remarks

By examining the response of land hydroclimate to two types of geoengineering approaches, we show that Sulfur Injection will lead to land wetting when the metric in use is P/PET, instead of P. This provides a counterargument to previous notions that Sulfur Injection will lead to over-drying of the land.

An additional promising feature is that Sulfur Injection can lead to a more rapid response in the next few decades, while the
benefit is only gradually getting stronger in the case of Carbon Capture despite an earlier effort to lower the emission. Even though the mitigation of the two approaches are both introduced in 2020 in these two experiments, the short-term (next 30 years) effect is stronger in the case of Sulfur Injection migration. The response time difference is due to the short-lived nature of sulfate aerosols, which will respond to mitigation measures more rapidly than long-lived species of CO2.

However, we also point out the weaker efficiency of reverting the Amazon drying due to the precipitation shift away from deep tropics to extratropic in response to Sulfur Injection. Moreover, we show that at a per-degree-cooling basis, Sulfur Injection is less effective than Carbon Capture to reverting the drying tendency globally and more so regionally over the Amazon (Figure 8, bottom row), because of the stronger precipitation suppression.

An important note should be made regarding the interpretation of the quantitative "benefits" presented. Because these two models have different climate sensitivities, the baseline warming induced by unchecked emissions growth is not at the same level. The WACCM model warms faster, reaches the 2ºC level at earlier decades, and has higher end-of-century warming at 6ºC, compared to the CESM. Since the baseline warming is different for the two models at different decades, the climate "benefit" due to any mitigation measures shall also be interpreted in a relative sense, i.e., the fraction (%) of the projected
change in the future that can be mitigated by the Carbon Capture or Sulfur Injection. For example, even if our results suggest Sulfur Injection can lead to 6ºC cooling while the Carbon Capture can lead to a 4ºC cooling, that does not quantitatively provide any constraints onto the strength of respective approaches.

We further note that the strength of each geoengineering approach, even when expressed in relative terms, is subject to the
assumption of applied forcing/perturbation. For example, the climate "benefit" can be enhanced by applying a larger Sulfur Injection or deploying Carbon Capture by a greater amount or earlier. In fact, in an earlier project that conducted a similar



Carbon Capture experiments (Sanderson et al., 2017), a weaker version of decarbonization was presented (by applying a smaller amount of emission cut and Carbon Capture), under which the temperature temporarily exceed the 2ºC level during the 21st century (i.e., "overshoot"), before falling back onto a stabilized 1.5ºC level.

As a result, we emphasize that the main purpose of this paper is not to examine the effectiveness of these two climate engineering schemes in the sense of absolute values. Instead, we aim to highlight the physical mechanisms at play, especially when distinct between the two approaches (e.g., radiative balances and dynamic response). The analytical framework established here will thus (1) help to understand the contrasting response in terms of spatial and temporal distribution, that goes beyond the specific regions highlighted here, and (2) provide broader insights to the mitigation impact of other geoengineering approaches beyond the two discussed here (such as cirrus ice cloud thinning or marine warm cloud brightening, Muri et al., 2018; and surface albedo modifications, Crook et al., 2015).

The quantitative results here using a suite of climate models thus need to be interpreted through the lens of model diversity. A recent example of such an attempt using two global climate models (Laakso et al., in review) focused on global mean precipitation response to carbon extraction and sulfur injection. Some aspects of the simulated responses here (e.g., the weaker sensitivity of precipitation to Sulfur Injection in WACCM) are worth revisiting by conducting systematic experiments using other climate models, such as the newly available CESM2(WACCM6). To place the two types of geoengineering schemes in a broader context, future research is also needed to further examine the upcoming model output from the GeoMIP6 (The Geoengineering Model Intercomparison Project Phase 6) and CDRMIP.

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





# Tables

**Table 1. The model experiments.**

| Experiment short name | Model | Simulation period | Ensemble size | Reference |
|---|---|---|---|---|
| CESM_Historical | CESM1-CAM5 | 1975-2005 | 30 | Kay et al., 2015 |
| CESM_RCP8.5 | CESM1-CAM5 | 2006-2099 | 30 | Kay et al., 2015 |
| CESM_CarbonCapture | CESM1-CAM5 | 2006-2099 | 10 | Sanderson et al., 2017 |
| WACCM_RCP8.5 | CESM1-WACCM | 2010-2099 | 20 before 2030, 3 after 2030 | Tilmes et al., 2018 |
| WACCM_SulfurInjection | CESM1-WACCM | 2020-2099 | 20 | Tilmes et al., 2018 |



'25 **Table 2. The changes in key hydroclimate variables (adjusted based on observation) over Major Land (60ºS to 60ºN). All numbers are derived from the time series in Figure 3. All changes are statistically significant at the 95% level. The definition of "major land" is the land regions over 60ºS to 60ºN, thus excluding cold regions where the seasonal snow cover or permanent ice sheet surface makes P/PET less useful as a predictor for aridity and vegetation types. The focused periods are Present-day: 2010-2019; Mid-century: 2046-2055; End of century: 2086-2095.**

'30

| Variables | T (ºC) | | P (mm/year) | | PET (mm/year) | | P/PET (unitless) | |
|---|---|---|---|---|---|---|---|---|
| Time periods | Mid-century | End of century | Mid-century | End of century | Mid-century | End of century | Mid-century | End of century |
| Changes under RCP8.5 by CESM, relative to present-day | 2.0 | 4.5 | 23.0 (2.9%) | 59.2 (7.6%) | 93.2 (7.5%) | 229.1 (18.5%) | -0.03 (-4.2%) | -0.06 (-9.2%) |
| Changes under RCP8.5 by WACCM, relative to present-day | 2.5 | 5.8 | 54.2 (7.0%) | 123.2 (15.8%) | 103.8 (8.4%) | 249.4 (20.1%) | -0.01 (-1.3%) | -0.02 (-3.6%) |
| CarbonCapture benefits (relative to CESM_RCP8.5) | -1.3 | -4.0 | -7.1 (-0.4%) | -36.8 (-4.2%) | -55.4 (-3.9%) | -188.8 (-14.6%) | 0.02 (3.2%) | 0.06 (8.8%) |
| SulfurInjection benefits (relative to WACCM_RCP8.5) | -2.4 | -5.7 | -40.4 (-5.2%) | -121.0 (-15.5%) | -128.0 (-10.3%) | -289.5 (-23.3%) | 0.03 (5.1%) | 0.05 (7.2%) |



**Table 3. Dryland area (million km² ) for semi-arid/arid types. North America is 15ºN-60ºN and 230ºE-300ºE. South America is 60ºS-10ºN and 280ºE-330ºE.**

| semi-arid/arid (km²) | Global Major Land | North America | South America |
|---|---|---|---|
| CESM_Historical (1975-2005) | 20.4/15.0 | 3.3/0.8 | 1.8/0.5 |
| CESM_CarbonCapture (2086-2095) | 20.7/15.1 | 3.4/0.7 | 1.8/0.5 |
| CESM_RCP8.5 (2086-2095) | 22.9/16.6 | 3.9/1.2 | 2.1/0.6 |
| WACCM_SulfurInjection (2086-2095) | 21.8/13.5 | 3.2/0.6 | 1.7/0.5 |
| WACCM_RCP8.5 (2086-2095) | 24.3/15.5 | 4.0/1.3 | 1.8/0.8 |

'35





'40

**Table 4. Global major land changes in P, PET, and P/PET, normalized by global temperature change. Two methods are used here: The first method is comparing the epoch difference of the end-of-century climate to obtain the normalized response. The second method is conducting linear regression over the decadal time series of anomaly induced by Carbon Capture or Sulfur Injection. In the rightmost columns, we contrast the results with the response to atmospheric CO2, tropospheric SO4 in the 20th century, as well as the various forcing experiments in the Last Millennium Ensemble (LME).**

| P (%/ºC) | Carbon Capture | Sulfur Injection | CO2 (Lin et al., 2016a) | Trop. SO2 (Lin et al., 2016a) | LME GHG | LME O3&AER | LME Volcanic 1250-1270/ 1450-1460/ 1800-1820 |
|---|---|---|---|---|---|---|---|
| Epoch difference | 1.1 | 2.4 | 1.2 | 6.7 | 1.4 | 8.1 | 2.6/4.1/ 2.5 |
| Linear regression | 1.1 | 2.7 | n/a | n/a | n/a | n/a | n/a |

| PET (%/ºC) | Carbon Capture | Sulfur Injection | CO2 (Lin et al., 2016a) | Trop. SO2 (Lin et al., 2016a) | LME GHG | LME O3&AER | LME Volcanic 1250-1270/ 1450-1460/ 1800-1820 |
|---|---|---|---|---|---|---|---|
| Epoch difference | 3.5 | 4.0 | 4.6 | 6.3 | 4.2 | 7.7 | 7.2 / 6.4 / 5.6 |
| Linear regression | 3.8 | 4.0 | n/a | n/a | n/a | n/a | n/a |

| P/PET (%/ºC) | Carbon Capture | Sulfur Injection | CO2 (Lin et al., 2016a) | Trop. SO2 (Lin et al., 2016a) | LME GHG | LME O3&AER | LME Volcanic 1250-1270/ 1450-1460/ 1800-1820 |
|---|---|---|---|---|---|---|---|
| Epoch difference | -2.0 | -1.5 | -3.3 | 0.4 | -2.7 | 0.3 | -4.9/-2.5 /-3.2 |
| Linear regression | -2.2 | -1.1 | n/a | n/a | n/a | n/a | n/a |

'45





**Table 5. Change (%/°C) in PET, and P/PET due to contributions of other climate variables (RH2M, U10, T, Rn-G, P). The variables used for this table are not bias-corrected because of the lack of reliable benchmark for U10 and Rn-G (surface available energy, which is the incoming radiation reaching the ground minus the heat flux into the ground).**

'50

(a) Contribution to PET change

| Carbon Capture / Sulfur Injection (%/°C) | RH2M (relative humidity at 2 meter) | U10 (surface wind at 10 meter) | T | Rn-G (surface available energy) |
|---|---|---|---|---|
| Global major land | 0.7 / 0.4 | 0.1 / 0.2 | 3.2 / 3.0 | 1.0 / 1.4 |
| North Brazil | 1.4 / 0.4 | 0.3 / 0.0 | 2.3 / 1.6 | 2.2 / 1.6 |
| South Brazil | 0.8 / 1.2 | 0.0 / 0.6 | 2.3 / 2.3 | 1.1 / 0.8 |

(b) Contribution to P/PET change

| Carbon Capture / Sulfur Injection (%/°C) | RH2M (relative humidity) | U10 (surface wind) | T | Rn-G (surface available energy) | P |
|---|---|---|---|---|---|
| Global major land | -0.7 / -0.4 | -0.1 / -0.2 | -2.7 / -2.6 | -0.8 / -1.3 | 1.7 / 3.0 |
| North Brazil | -1.1 / -0.3 | -0.2 / 0.0 | -1.5 / -1.2 | -1.6 / -1.3 | **-3.4** / 0.9 |
| South Brazil | -0.7 / -1.1 | 0.0 / -0.4 | -1.9 / -1.8 | -1.0 / -0.7 | 1.6 / **-1.0** |

'55



# Figures

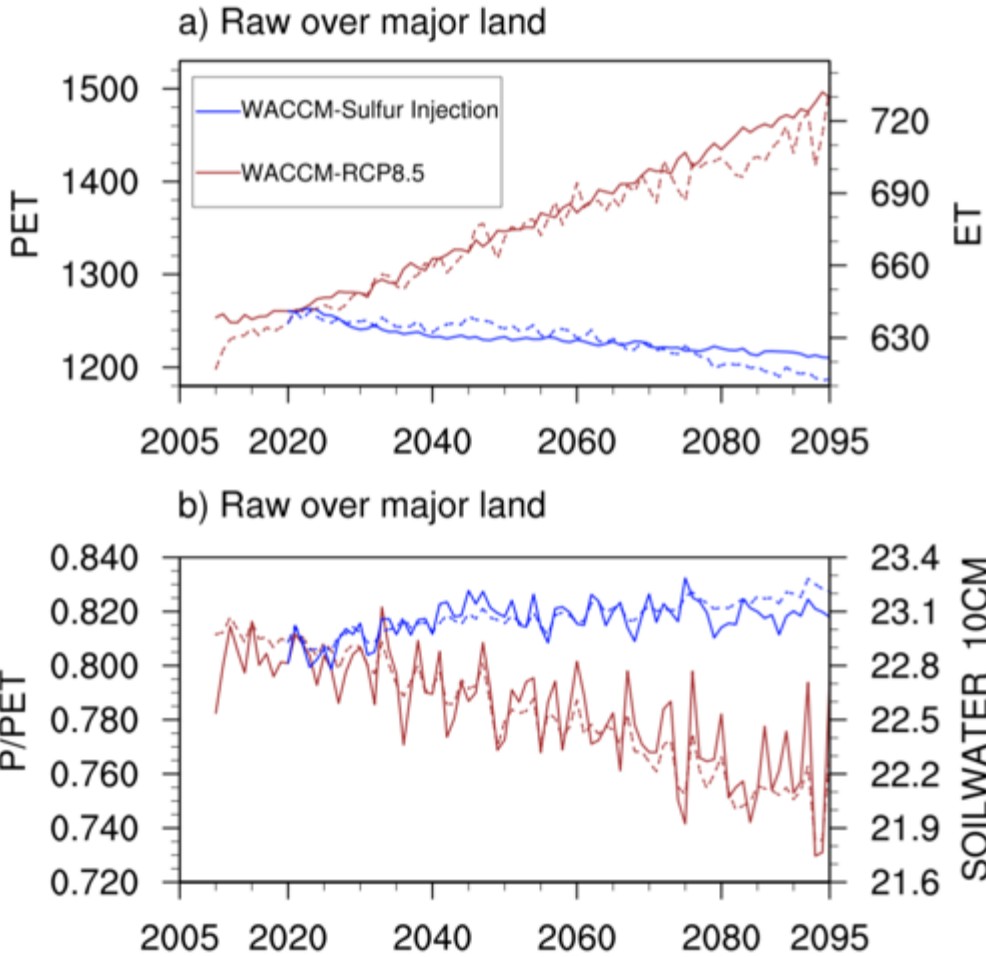

'60

**Figure 1. (a) PET (mm/year) and ET (mm/year) as simulated by WACCM, over global major land regions (60ºS to 60ºN) only, because P/PET is not a good indicator of drought condition over snow-covered cold regions. (b) P/PET (unitless) and soil water in the top 10cm layer (mm). The solid lines are for the left Y-axis and the dashed lines are for the right Y-axis. Red lines are under RCP8.5 and blue lines are under Sulfur Injection. Note that no bias correction or smoothing was performed for this figure. CESM**

'65 **results are not shown because (a) there is no reduction of soil moisture in the RCP8.5 warming and (b) there is no separation of soil moisture in the long-term trends between the RCP8.5 and mitigated warming simulations, presumably due to model deficiency outside the Americas, because within Americas (Figure 2), the CESM results are largely consistent with WACCM.**





'70

'75

**Figure 2. (Left two columns) WACCM simulated P/PET (unitless), soil water (mm), P-ET (mm/day), at the present day, and its end-of-century change under RCP8.5. (Right two columns) Same as left, but for CESM. Note that all data here in Figure 2 are without bias correction as in Figure 1. The focused periods are Present-day (2010-2019) and End of century (2086-2095). The dashed regions in the panels of "End-of-Century – Present-day" are where the differences are significant. Similar results are presented in Figure 5 but after bias correction and in relative terms (%).**

'80

'85

**Figure 3. The annual average of major land temperature (T) (°C) (the first row without bias correction and the second row with bias correction), and major land precipitation (P) (mm/year) (the first row without bias correction and the second row with bias correction). The third row shows the major land PET (mm/year), and major land P/PET (unitless) (after bias correction). Shading represents the standard deviation of ensemble members. We have applied decadal smoothing to all the time series but note that WACCM-RCP8.5 has 3 ensemble members during 2030-2099, and thus has large fluctuation due to natural variation, in contrast with 10 or more ensemble members in other cases.**

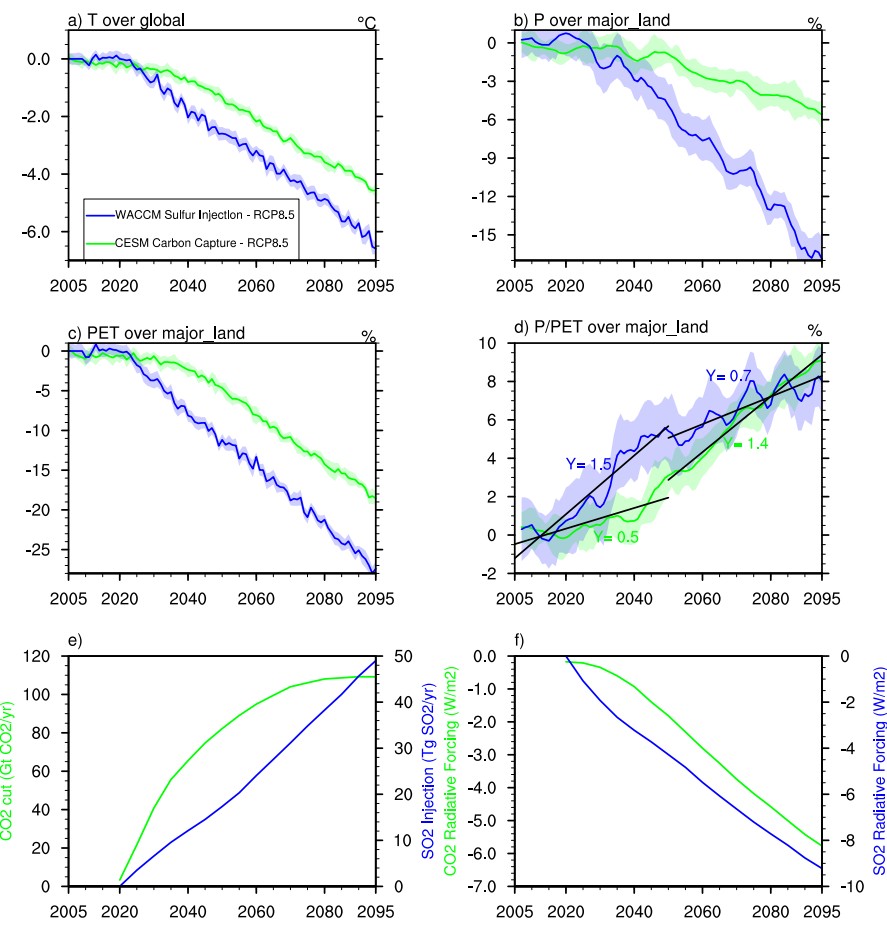

'90

**Figure 4. The changes of T, P, PET and P/PET due to two types of geoengineering schemes. The changes are shown in relative terms (except for T in ºC), which is computed from the absolute values in Figure 3. In panel (d), the linear fit for the blue and green lines (separately for 2005-2050 and 2050-2095) is to highlight the near-term "benefit" of Sulfur Injection. (e) the mass of carbon capture (Gt CO2/yr) and sulfur injection (Tg SO2/yr). (f) the radiative forcing (W/m2) due to carbon capture and sulfate injection. The mass**

'95  **of carbon capture and corresponding radiative forcing are based on Sanderson et al. (2017) Fig. 1. The mass of sulfur injection comes from Tilmes et al. (2018) Fig. 2, and the sulfate radiative forcing is calculated based on the equations in Metzner et al. (2012).**





**Figure 5. P/PET in the present day, the changes in the future under RCP8.5 and mitigated due to two types of geoengineering.**

**The first row: the reanalysis and the bias-corrected output of the two models for the present day (unitless).**

**The second row: (d) The projected baseline changes using CESM at the mid-century in the relative term (%); (e) The mitigation of the baseline changes due to Carbon Capture; (f) The mitigation of the baseline changes due to Sulfur Injection.**





**The third row: same as the second but for the end of the century.**

**The similar changes of P/PET in absolute values (unitless) and WACCM results (but without bias correction) can be found in row 1 of Figure 2.**





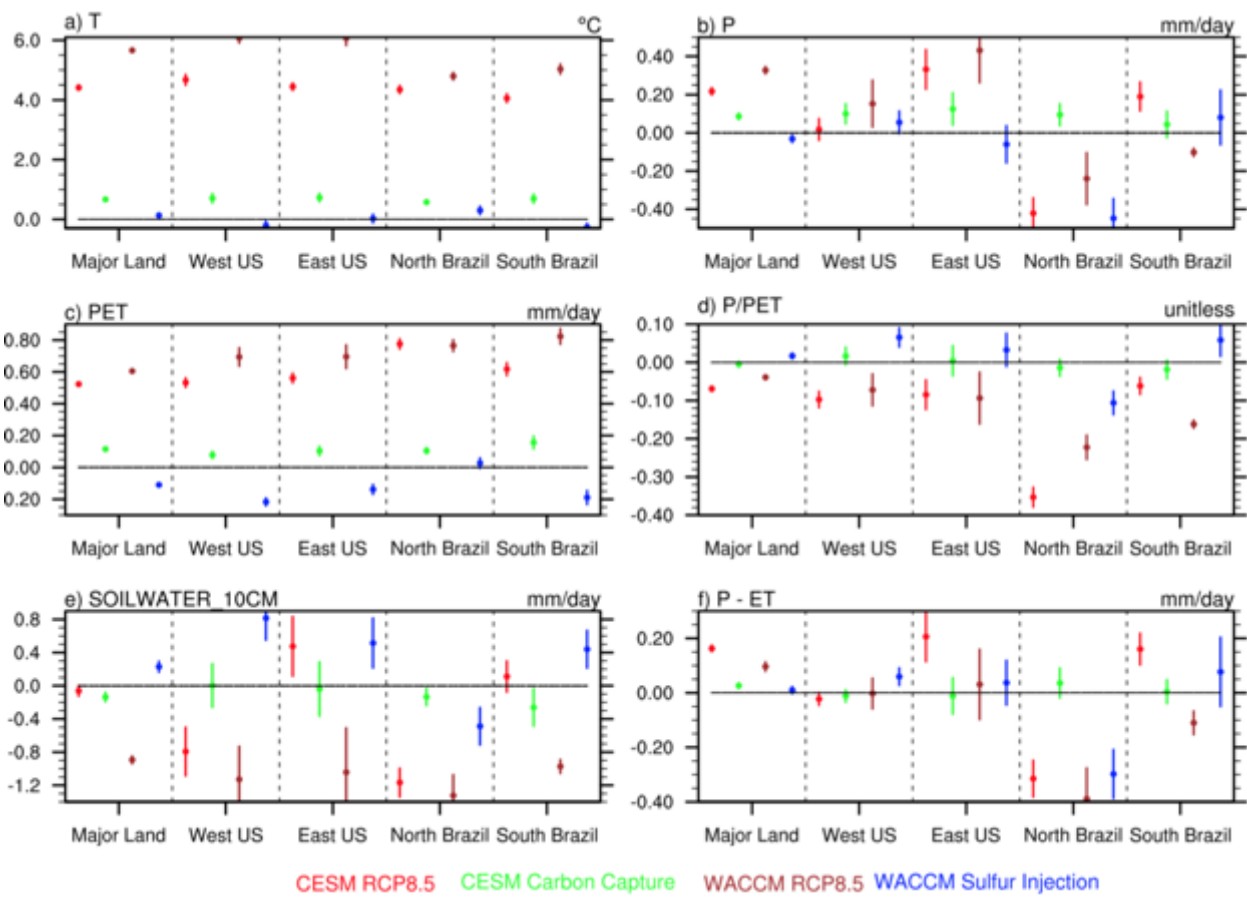

**Figure 6. Regional mean changes at the end of the century in terms of the anomaly relative to present-day. Western and Eastern US are separated by 100ºW, and Northern and Southern Brazil are separated by 10ºS. See the boxes in Figure 5a for the domains. The zero line of y-axis indicate the present-day level.**



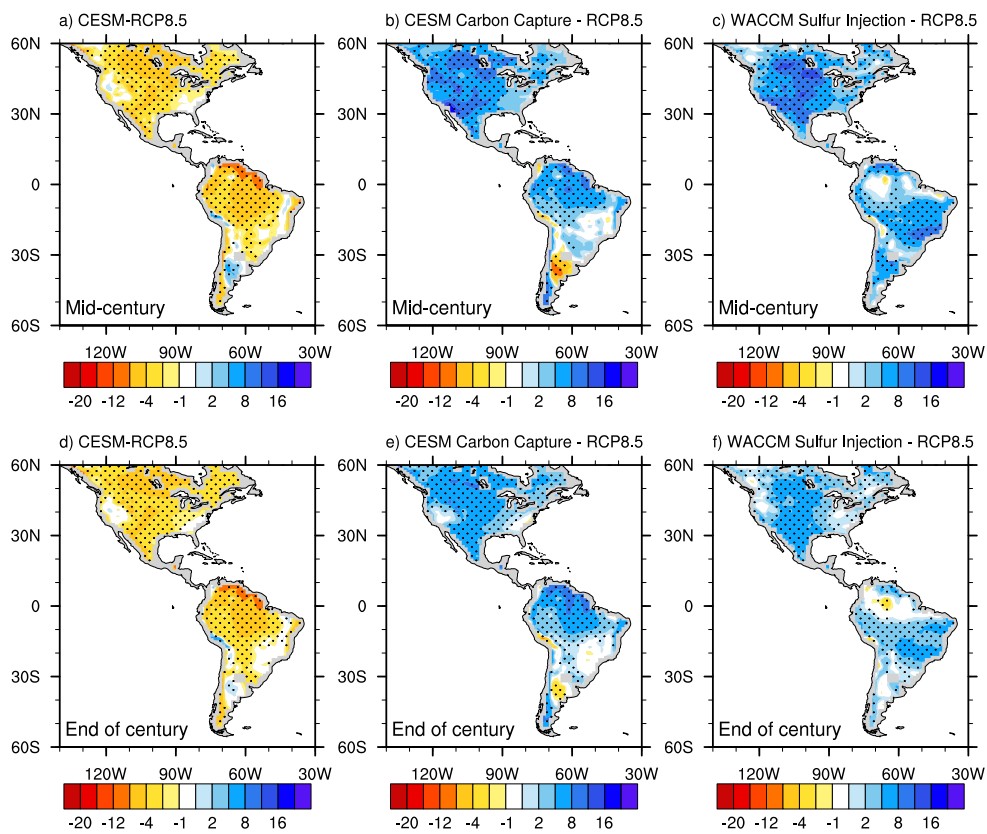

515

**Figure 7. Similar to Figure 5's 2nd and 3rd rows but showing normalized P/PET change in %/°C. (a)-(c) are mid-century changes; and (d)-(f) are end of century changes. The normalization is by scaling the global temperature change in the right two columns, for example, 4.2°C due to Carbon Capture in panel e and 6.0°C due to Sulfur Injection in panel f.**

520





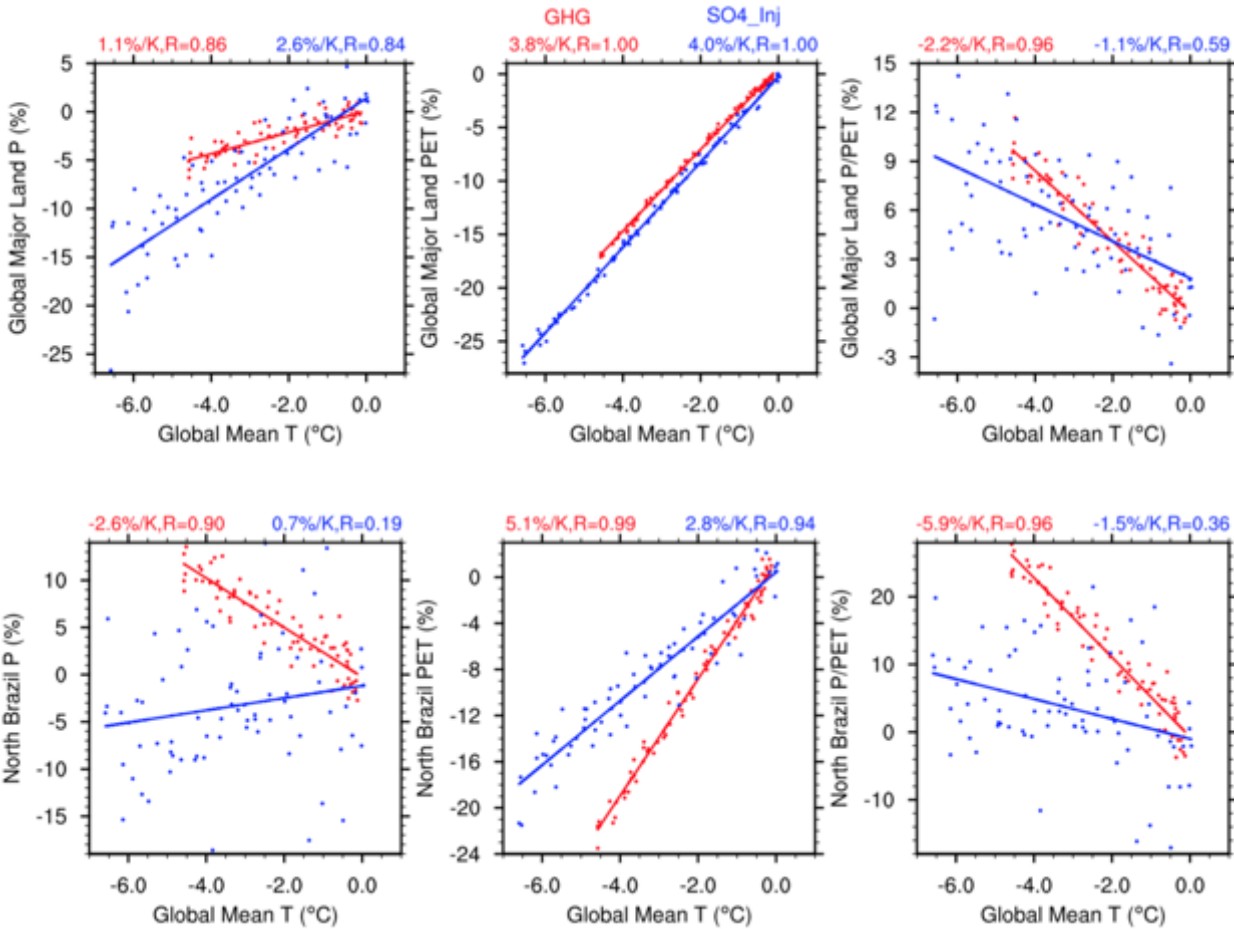

**Figure 8. The changes of global major land P, PET, and P/PET as a function of global mean temperature change (ocean+land). Each dot represents a model year during 2020-2097. Red is for Carbon Capture, and blue is for Sulfur Injection. (top) global land; (bottom) North Brazil.**



**Figure 9. Similar to Figure 7 but showing (Row 1) RH2M (relative humidity, normalized change in %/°C). (Row 2) U10 (surface wind, normalized change in %/°C). (Row 3) Rn-G (surface available energy, normalized change in %/°C). (Row 4) P (precipitation, normalized change in %/°C). Row a (left) is the changes in the baseline warming. Row (b) is the mitigated change due to carbon capture. Row (c) is the mitigated change due to sulfur injection. Note that we show RH2M and U10 results for mid-century, and Rn-G and P results for the end of century.**