# Peer review of "Climate engineering to mitigate the projected 21st-century terrestrial drying of the Americas"

_Earth System Dynamics, 2020_

## Referee Comment (RC1) · Anonymous Referee #1 · 24 Feb 2020

The author studied the climate response to carbon capture and stratospheric sulfur injection geoengineering, using simulation results from two coupled global climate models. The author focused on the climate response, including temperature, precipitation, and aridity over major land area especially North and Southern America. While the scientific question examined here could further our understandings of the potential impact of different geoengineering strategies on the regional climate response, this paper fails to convince me to be useful in a number of aspects (Discussed below). Therefore, I don't think the current paper is suitable for publishing unless a substantially improved version is made.

[Figure]

First of all, the way the paper is written could be improved substantially. Many important details that are crucial for readers to understand their results are missing, which confuse me a lot during my reading of the paper. For example, in the Methods section, the model and experiment descriptions are just too simple. The author compares simulation results from two coupled global climate models, CESM-WACCM and CESM-CAM5, but how different these two models are, and how these differences could affect simulated climate response is rarely included. Also, the author mentioned that the land component differs from each other for these two models, but did not include any discussion about what the difference is. The description of the experiment design and post-processes is also not satisfied. For experiments used in this study, did you use the GLENS project results for the Sulfur Injection case? Where did you get the Carbon Capture simulation results and what is the trajectory of the CO2 concentration/emission in this simulation? For the bias correction, I am confused by how exactly the bias correction is applied to the future simulation cases to account for different climate responses for these two models? etc. Another example is that in section 2.2, the authors mentioned "CO2 capture and massive CO2 emissions cuts applied to start in 2015-2020", in the next section the author says "both schemes are designed to be deployed in 2020", I am confused which one is accurate? The author should at least state such basic information clearly in the paper.

I am also not convinced by the paper. In many places, the author compared the difference of climate responses to two geoengineering scenarios. However, these scenarios are simulated using two different global climate models, and the authors have mentioned that climate sensitivity differs substantially for these two models. How should I believe the difference in climate responses is due to different impacts on these geoengineering schemes, not just because of the difference in the structure of these models? Section 5 is helpful in some way, but I am still not sure how should I interpret most of these results? Many descriptions presented in the paper seem not accurate to me. For example, the authors claim that "spatial patterns show similar agreement between P/PET and soil moisture as well as P-ET" in figure 2. While the time series do show

similarities between different indexes, the spatial patterns show opposite responses in many regions, especially Eastern North America and Southern South America. I am not sure if this could be seen as similar?

Line 7, the authors claim that "it is essential to apply the bias correction to model output before carrying out a meaningful comparison of future changes". If you are using simulation results from the CESM_RCP8.5 2010-2019 period to correct both WACCM_RCP8.5 and WACCM_SulfurInjection, and since WACCM_SulfurInjection is the same to WACCM_RCP8.5 before 2020, why should the difference between these two cases change compared to results without bias correction? Can you compare these simulations using the same model for all these simulations? Otherwise, I am not sure what is the best way to account for difference caused by using different models.

Line 10-12: First of all, I think the author should emphasize more that the analysis here is only for changes over land, especially when you are presenting the results. The author claims that the larger warming over land for WACCM is due to the larger climate sensitivity of the model. The two models differ not only in climate sensitivity but also the ratio of the land to ocean temperature responses. How much of this larger warming is due to larger climate sensitivity, and how much is due to different land/ocean temperature responses? The same concern applies to the next paragraph (lines 16 to 20). How much of the larger warming in WACCM compared to other models is due to climate sensitivity or land/ocean responses in these models? If the main reason is climate sensitivity, then the larger cooling in WACCM could just because WACCM produces more warming and the goal is to stabilize global mean temperature changes at the same level. I don't think the reason is simply that a larger injection amount or the injection strategy. If the main reason is the different land responses, then does the explanation in the previous paragraph using climate sensitivity still hold? Also, I don't know what you mean by saying "the longer lifetime of $CO_2$ makes the impact of perturbation to the carbon cycle and atmospheric concentration slower to emerge". Factors that affect the cooling effect of $CO_2$ capture include the inertia of the ocean,

the response of the global carbon cycle, etc. I don't know what does the longer Co2 lifetime means here. The last sentence point to Table 2, column 3. I don't know where to find these results? Also, the author saying "if only the land variables are considered". Do you have any explanation for this?

Line 54 to 56: From the time series plot, you showed a decrease in precipitation by comparing results between WACCM_SulfurInjection (2086-2095) - WACCM_SulfurInjection (2010-2019), but in table 2, the decrease in precipitation for end of century is 121.0 mm/day compared to WACCM_RCP8.5, smaller than the increase in WACCM_RCP8.5 compared to present day (123.2 mm/day) value. If I understanding this correctly, the present-day value should be the same for WACCM_RCP8.5 and WACCM_SulfurInjection because they branched from each other in 2020. So why is there a decrease instead of a slight increase in precipitation (am I reading this wrong here)?

Line 102-104: It is clear from Fig.4d that the P/PET index has different changing rates before and after 2050 and since changes in PET is quite consistent throughout the simulation, the difference is mainly caused by different responses in P (Maybe you should do the same calculation and see how much these two terms contribute to the difference in P/PET). However, the authors attribute these differences in P simply to "when the surface cooling starts to emerge, the precipitation also weakens in response to Sulfur Injection". What do you mean by saying "the surface cooling starts to emerge"? The changes in P/PET results seem interesting to me, but I don't think the authors explain these changes in an appropriate way here. I don't even understand what you mean by saying "when the surface cooling starts to emerge" here?

Section 5 investigated the contribution of P and PET to the change in P/PET for different scenarios. For the discussion of P, you compared the precipitation results of stratospheric sulfur injection with tropospheric SO2 increase, and then say "the larger slope of precipitation for Sulfur Injection is expected." Could you directly compare these two results under different forcing designs? I am not sure how could you then get the

conclusion that "the model dependence is the largest uncertainty" when you are not comparing the same kind of forcings? Also, for the regression results, did you check the standard deviation for these regression results? How robust the regression results are? It is interesting that global major land PET scales almost linearly with global mean T. If the PET is such tightly related to the T change, why is there a difference in PET slope between these two forcings? I don't think the authors explain it in a reasonable way. If you attribute this also to model difference, then maybe you should use the same model to do the comparison.

---

## Referee Comment (RC2) · Anonymous Referee #2 · 24 Feb 2020

Xu et al. compare the changes in aridity under a worst-case warming scenario (RCP8.5), an extreme mitigation and carbon dioxide removal scenario (that I'll refer to as CDR) and the GLENS sulphate aerosol injection scenario (that I'll refer to as GLENS). They combine results from 2 versions of the same base model (with importantly different climatologies) CESM1, which ran the CDR scenario, and WACCM, which ran the GLENS scenario, to assess the surface hydrological response to these different scenarios. To make the two ensembles comparable bias-correction is applied such that both models are adjusted to match the observed climatology. The authors evaluate a range of hydrological measures and settle on P/PET as their choice of aridity measure. Focusing on this metric they evaluate changes over the global land area (I'm

guessing this is what is referred to as "major land") and across the Americas, and evaluate the differences between the CDR and GLENS response. They also normalize the response by global temperature to address a mismatch between the CDR and GLENS experimental objectives. They find that RCP8.5 produces a general aridification over the land and over the sub-regions they focus upon, driven by a large increase in PET with regionally divergent P responses. They find that while CDR generally partially offsets these trends, GLENS goes too far producing a substantial net reduction in PET that more than offsets the reductions in precipitation. The Amazon is an exception for GLENS, where instead of producing a net increase in P/PET instead it only partially offsets the substantial reduction seen in RCP8.5.

General comments

My main reservation about this paper is that I don't believe comparing these scenarios is policy-relevant or that scientifically useful. The title sets this up as a competition (vs.) between two geoengineering options but these policies are complementary. The CDR scenario (carbon capture as it's often referred to) is also not primarily a CDR scenario but largely an emissions reduction scenario as compared to RCP 8.5 with net negative emissions only after 2050. The other issue is that the experiments were designed to achieve different objectives and produce quite different radiative forcing and temperature responses over the analyzed period and were run in model-versions that differ in quite important ways. Both of these factors undermine the utility of this comparison. That said, clearly a lot of careful analysis has been done and so I'll continue to my other comments.

Much of the paper is written as if the two experiments had the same aim but they didn't. The methodology doesn't make the different goals of the two experiments clear enough and their high-level difference ought to be described up front, i.e. section 2.2 should be elaborated upon to make clearer the experimental setups and their differences. There is often reference to "lagging effects" of the CDR experiment but given that it had different ends from those of the GLENS experiments it seems inappropriate to

describe the temporal evolution as if it were trying to achieve the same thing.

I was surprised that transpiration wasn't discussed as this is a major driver of terrestrial hydrology. I strongly recommend considering the direct $CO_2$ physiological effect of $CO_2$ in these simulations as this will be a big difference between CDR and RCP8.5 / GLENS. The authors report results showing PET and ET over land, finding that they both increase in lockstep. This was surprising to me, given the results of Swann et al. (2018) who found that PET was projected to increase in models while ET was not. Swann et al. found that this was due to reductions in transpiration due to $CO_2$ fertilization offsetting the meteorological drivers that drove PET increases. This suggests that the $CO_2$ physiological effect has not been included or is very weak. Please provide details and analysis to address this. Relatedly, it is not clear whether there were any differences in land cover between the CDR scenario and the RCP8.5 / GLENS scenarios. This could have a large effect on regional hydrology.

The temperature difference between the two experiments should be made clear, from table 2 I can back out that CESM-1.5C is 0.7C / 0.5C warmer than the baseline whereas GLENS is 0.1C warmer (is this right? I thought it was designed to perfectly offset warming from the baseline period). Given the lower sensitivity of CESM, this 0.5C figure would be larger if both were run in WACCM. This could be driving some of the difference between these two experiments and should be brought out and discussed. I'd like a clearer sense of the magnitude of the bias-correction overall and of the regional character of it in your study region. The fact that WACCM has around 25% more precipitation over "major land" (undefined as far as I can tell) ought to be highlighted! Beyond this the regional biases ought to be made clearer. Are the models far off in key regions such as amazon? How wrong are the arid and semi-arid areas calculated using the models uncorrected P/PET values? How different are the regional biases? An extra figure or two is needed to make this clear.

The temperature normalization section doesn't seem to add much to the paper and has some serious problems. I understand that it could be useful for comparing scenarios

with different levels of cooling if it were not for the difference in the climate sensitivity. WACCM's climate sensitivity looks to be ∼20% higher than CESM's which will mess with this normalization procedure. I'd suggest either cutting this section (what do we learn that isn't covered elsewhere?) or addressing this climate sensitivity issue by testing how different the RCP8.5 sensitivities are between these 2 models. I worry that model differences rather than scenario differences could be driving some of the response seen in this section. The section summaries seem unnecessary.

I think it would be more fair to describe the "carbon capture and storage" scenario as an extreme mitigation scenario. A reduction in positive emissions makes up the bulk of the difference between RCP8.5 and this scenario. If you wish to highlight the use of carbon dioxide removal (CDR) or the presence of negative emissions then I'd suggest using these terms instead of carbon capture and storage as this technology can be used without producing negative emissions, i.e. on coal power plants. I'd suggest coming up with some clear shorthand for the experiments and using it consistently in both text and figures, e.g. CESM-RCP8.5, CESM-CDR, WACCM-RCP8.5, WACCM-GLENS, Baseline. The figures have inconsistent labeling, line colours and styles and some of the captions are oddly formatted.

The manuscript text needs a careful proof-read by a native English speaker. There were too many grammar mistakes so I only addressed the worst. There were also many very short paragraphs that could be merged with their neighbors.

Specific Comments

N.B. Specific comments are given in the order that they appear in the manuscript with the line numbers as I saw them. It seems the pdf has cut off the hundreds part of the line number and I haven't converted the cycling line numbers into something more sensible.

Title – Given that you stress in the conclusion that you aren't trying to evaluate which is better and instead are focused on the mechanisms, I'd suggest: "climate engineering and aridity in the Americas: a comparison of carbon dioxide removal and sulphate aerosol injection" 22 – I'd argue that this paper hasn't used a consistent framework: different models and different objectives for deployment 22-24 - Given that this is more a mitigation scenario than a pure-CDR scenario, it's not correct to describe this as the first paper to compare sulfate geoengineering against "carbon capture" as previous studies have made such comparisons: [Niemeier et al. 2013: DOI:10.1002/2013JD020445; Muri et al. 2018; DOI:10.1175/JCLI-D-17-0620.1; Jones et al. 2018, DOI: 10.1002/2017EF000720, etc.]. 32 – what does "mitigation potential" mean in this context? 39 – does it worsen the trend or is the trend worsening under RCP8.5? 67 – Vaughan and lenton 2011 don't provide evidence of investment, do they? 68 – I thought CDR installations would be effectively independent of emission sources. CCS is installed directly onto power plants, etc. but that's classed differently. 89 – worth comparing that to the "dry-gets-drier" pattern of global warming Sect. 2.1 This is a very short model description, and includes only one citation to one of the model versions used. Please elaborate. Sect 2.2 Both experiments need to be described in more detail. 30 – "to stabilize TEMPERATURE at 2020 levels..." would be more accurate. 35-41 – Is this based on another RCP, How large are the negative emissions, are there any differences in the land surface (more forest cover, etc.)? This description leaves out crucial details. Please outline them here even though they are explained in detail in the Sanderson paper. Sect 2.3 – This is too little material for a sub-section, so I'd suggest cutting it or else expand it to discuss more of the analysis approaches employed, e.g. the time-periods covered and the procedure to normalize by global temperature change that is employed later. 65-59 – This seems out of order, I'd suggest moving figures 1 and 2 out of the methods section and into the results section. Figure 1 – It would be much clearer to color one axis red and the other blue and use dashed versus normal to separate the experiments, then one could read the figure at a glance rather than having to get half-way through the description to know what is shown. Figure 1 – This shows ET and PET increasing in lock-step but Swann et al. 2016 (www.pnas.org/cgi/doi/10.1073/pnas.1604581113) showed all climate models they investigated diverging with PET rising rapidly and evaporation not rising at all. Swann et al. argued that this was due to the direct physiological effect of $CO_2$ suppressing transpiration. That would suggest that there is no direct physiological effect in your simulations, is that correct? If not, can you explain the difference between your results and those of Swann? Figure 2 – "significant" – please elaborate here or in the methods. L80-84 – These descriptions are incomplete. I cannot tell whether this is a reasonable approach as the terms are not defined. 89-99 – This paragraph is hard to follow, I'd suggest revising it. 89-91 – Is the same bias correction applied to both models or are the separate historical bias corrections applied to the future in both models. 93-94 – I don't believe this is the same "present-day" as in the GLENS experiments which I believe aim to keep conditions fixed at their 2010-2030 levels. Section 3.1 – The sub-section title seems to be at odds with the section title, perhaps change the section title? Figure 3 – What is "major land"? Fgiure 3 – The change in axis range between b and d should be avoided. You should make clear visually or in the text that WACCM's precipitation over "major_land" is ∼25% too high. This is huge! 10-11 – What is WACCM's climate sensitivity? If it's not known then I'd reverse the order and highlight that it's higher than the already-high 4 C of CESM. 13-14 – This is the first time that this has been mentioned, I'd suggest highlighting this fact in the methods section. 16-20 – The off-tropics injection is irrelevant as if the sulphates were less effective more would have been injected. Of course there's also the afore-mentioned high climate sensitivity, which should be mentioned. 22 - "global major land" needs to be defined. 25-28 – Isn't it better to describe this as a fundamentally different experiment given their different aims? The carbon capture and GLENS experiments have different ends that end up producing a roughly similar temperature response, i.e. little change from 2020. Table 2 – Seems unnecessary, suggest cutting. You've already shown this in figure 3, and reported many (perhaps too many) of these figures in the text. 29-31 – These lines are unclear, larger than what? Also unclear what is being referred to. Is column3, mid-century P or is it all of the PET results? Do the variables count as a column? 35 – I'd avoid the term "mitigation" given its use to refer to emissions cuts in

the climate literature. I'd suggest renaming this "mitigation potential" to something else. 37 – explain that this is a reduction of a precipitation increase not a 100% reduction in precipitation. 38-63 – These paragraphs on the global-mean precipitation response ought to be revised, they are not well written. It is also strange to refer to an almost perfect reversal of RCP8.5 precipitation trends as being over-effective. L63 – Could you state what "this precipitation-centred argument" is or else reframe this. L67 – I'd avoid referring to sulfate injection as a mitigation scheme. L67 – why aspire to do it and not just get on with it? 70 – does it? I see a drying in all regions in P/PET and a wettening trend in most places except the amazon and central America. 73-75 – this sentence is mangled. 75-77 – is this in the RCP8.5 experiment? 77 – "bring forth the benefit of curbing aridity worsening" 89 – I'd leave this type of commentary until the discussion as its tangential to what is being described. 95-96 – Better to describe this as changing the sign of the trend as reversing is ambiguous in this context, i.e. it could mean simply offsetting. 98-99 – the shorter response time doesn't explain this. P/PET rises rapidly then stops rising, Why is this? Figure 4. Again, these experiments are so different in character it seems odd to compare their time evolution. They have a different temporal evolution of forcing (in ways not described or explored in this paper) so its unsurprising that they have different temporal evolutions of climate response. Section 3.4. I'd avoid the generic sub-section title "summary" and give something specific to this section. 10-14 – As I mentioned earlier, the results in figures 1 and 2 do not show the same pattern for the different variables so I don't think it fair to say P/PET stands in for all measures of aridity as this text implies. 18 – where is the CO2 captured results? 21 – This type of discussion should be saved for the discussion section. Section 4, how about "hydrological change in the Americas"? Figure 5. This figure has too much going on and should be split or else the mid-century results cut. The WACCM RCP8.5 results are missing which are crucial for interpreting the WACCM GLENS – RCP8.5 anomalies plotted. I'd suggest replacing the bias-corrected model results with the original model results. It is unsurprising that the bias-corrected results look very similar to the observations. The models simulated present day has a direct bearing on its projections for

the future whereas the bias-corrected present day doesn't. I'd also suggest making the CDR and GLENS anomalies relative to present-day rather than RCP8.5. Finally, The caption format shouldn't include paragraph breaks. 43-45 – which sees this to a lesser extent? Is the global value "major land"? 46-48 – this seems redundant Table 3. The formatting / caption could be clearer here. Is CESM historical the same as historical after bias-correction, are they similar? 51 – make clear that you are referring to the global arid area. 4.2. This whole South America section is poorly written, I'd suggest revisiting it. 61-62 – revise this sentence. 65 – I'd suggest using 2 significant figures here. What does the 15-30% refer to? 66-68 – Is the amazon really the only region in the Americas to see a decline in precipitation in RCP8.5? Figure 6 – Given the differences in the climates of the regions under investigation would it not be better to report results in percentage change terms? There's a similar ~50 mm/day increase in PET in West US and Brazil but presumably they have very different absolute values. 73-74 – this isn't a good description of what is in the figures, P is up in all regions in the CDR experiment which is just different from what is going on under RCP8.5 and around 80-90% of the PET response is offset. 79 – why not open with the big picture for GLENS then address this exceptional response? 83-85 – it's not consistent with Simpson et al if it's the same data, it's just the same thing. 94 – continues? When did it start? 96 – There's a larger reduction in PET in GLENS, so the differences should be due to the differences in the precipitation response. Should probably note that the precipitation reduction in CESM-RCP8.5 is around twice as large as in WACCM-RCP8.5. 00-01 – Which regional trend is being referred to here? 11-13 – Again, what is being referred to here? Section 4.3. Again, I'd avoid using "summary" for a sub-section title. 37 – it's not a further decrease in precipitation if the decrease in precipitation is less than in RCP8.5. 50 – again, I'd suggest avoiding using mitigation in this way: "mitigation capacity" 52 – Presumably WACCM-CDR (if you had run it) would be about 20% less effective than CESM-CDR by this measure as WACCM has a higher climate sensitivity and so the response would be divided by ~6C rather than ~5C. Figure 7. I don't see any value in including this figure. How is the statistical test applied here? 56-59 – This

is a result of not changing the scales of the plot to reflect the fact that you've divided it by ∼5. 62 – Higher climate sensitivity is the obvious driver of this but isn't mentioned. Figure 8. It is not clear what is going on in this plot. Which model is plotted? Are these the bias-corrected results or not? This plot needs to labeled more clearly. I'd suggest producing a similar plot for the RCP8.5 results of the 2 model versions as I suspect that they could look quite different and this might be driving some of the difference attributed to the forcing differences here. 85-87 – missing citation to earlier work. 90-91 – missing again. 95-97 – it could also be that tropospheric aerosols are concentrated over the land. 06-07 – rather than reversing, I'd recommend "changing the sign of" or some other construction that's less ambiguous. Table 5 – The formatting is not great, I don't think the X / Y format is the best choice. The simulated PET value should also be reported as well as a total or mismatch column. 18-22 – This paragraph should be revisited. 26 – this is not a counter-argument, P/PET is a common measure of aridity P is not. 29-33 – The experiments are very different, this paragraph describes them as if they had the same goals. 35 – I'd suggest "less effective at offsetting the amazon drying"? 40-47 – This difference in climate sensitivity also undermines the normalization procedure in section 5.

---

## Referee Comment (RC3) · Anonymous Referee #3 · 1 Mar 2020

In this study, the authors compared model-simulated hydrological cycle change in two scenarios: a scenario in which global mean temperature, equator-to-pole temperature gradient, and interhemispheric temperature gradient, are all stabilized at present day level under the RCP8.5 background scenario through stratospheric sulphate aerosol injection (GLENS ensemble simulations); and a scenario in which atmospheric $CO_2$ is reduced to achieve the temperature stabilization goal of 1.5 degree (carbon capture and storage). The stratospheric sulphate injection simulations are done with CESM-WACCM, and the carbon capture and storage simulations are done using CESM1-CAM5. The main metrics used in the analysis are precipitation (P), potential evapo-transpiration (PET), and the ratio of P to PET. The regions focused on is North and

[Figure]

South America.

My biggest concern is to what extend hydrological cycle change in these two scenarios can be compared with each other. By experiment design, compared to the present-day climate, the global mean temperature is near zero in the GELENS, and 1.5 degree warming in carbon capture and storage simulation. Different amount of temperature change would certainly be one of the major factors responsible for different hydrological cycle change. The authors also presented changes that are normalized by global mean temperature change, but to what extend these hydrological metrics, in particular PET, scales with global mean temperature at the regional scale? CESM-WACCM and CESM1-CAM5 also has different model configuration and climate sensitivity, which further complicates the comparison between two sets of simulations.

In the conclusion, the author states that 'As a result, we emphasize that the main purpose of this paper is not to examine the effectiveness of these two climate engineering schemes in the sense of absolute values. Instead, we aim to highlight the physical mechanisms at play, especially when distinct between the two approaches'. But most of the study is actually devoted to the comparison of these two scenarios quantitatively, and I really don't see a clear presentation of the fundamental physical mechanisms gained from this study.

Specific comments:

Abstract: 'these two leading geoengineering schemes have not been carefully examined under a consistent numerical modelling framework.'

Does it imply that this study is the first to carefully examine these two schemes in a consistent modelling framework? This is clearly not true.

'Here we present a comprehensive analysis of climate impacts . . .'

This is not true. This study only analyzes some hydrological metrics for some specific regions. This is not a comprehensive analysis. Introduction:

Line 64: The reference of Xu and Ramanathan, 2017 and Miller et al., 2017 is missing in the reference list. (please also check other references. Quite a few are missing in the reference list)

Line 64-65: Climate engineering is proposed as a potential method to mitigate global warming, but it is a too strong statement saying that climate engineering is needed in climate mitigation. In fact, in the abstract of Lawrence et al. (2018), as cited here, it states: "Based on present knowledge, climate geoengineering techniques cannot be relied on to significantly contribute to meeting the Paris Agreement temperature goals"

Line 65: Please provide reference for these approaches. In particular, what is 'spraying sea water over sea ice' and 'oceanic evaporation enhancement' approaches?

Line 67-70: It is confusing to state that they are global-scale schemes. In theory, each of the schemes described here can be implemented at either global or local scales.

Line 70: This sentence needs some rewriting. Stratospheric sulphate injection is usually considered to be relatively inexpensive.

Method:

In addition to fundamental difference between GLENS and CO2 mitigation, these two sets of simulations use different versions of CESM, which adds another uncertainty to the results presented here.

By just reading the paragraph of the carbon capture and storage experiment, it's not clear to me whether this is emission-driven or concentration driven. It says net emission is reached at year 2050, and then says the corresponding CO2 concentration is prescribed rather than simulated. Also, a figure showing the emission (or concentration) pathway for the carbon capture and storage experiment should be presented.

2.4 Hydroclimate variables examined: The authors state: "we focus on climate quantities over land due to their close relevance to agriculture, ecosystems, and the carbon cycle ..". Why not also analyse some variables directly related to agriculture and carbon

cycle, such as terrestrial gross and net primary production? They are available from CLM output.

Page 6, line 71: "Climate model output cannot be taken at its face value". This statement is not true. It depends on purpose. For climate modelling studies that aim to understand fundamental mechanisms, no bias correction is needed at all.

3. Mitigation at the global scale Most of this section is devoted to the presentation of numerical values and the characteristics for temporal evolutions of temperature, precipitation, and potential evapotranspiration. This kind of discussion should be shortened and replaced by scientific discussions of the underlying mechanisms.

It is not surprising that temperature response to SAI is quicker than that to carbon capture because of the long timescale associated with $CO_2$ forcing, but I think more analyse should be done for the change in hydrological cycle. How does the change in PET compare the change in soil moisture as presented by Cheng et al. (2019)?

For carbon capture simulation, what is the role of the direct $CO_2$ effect on land through the influence on stomatal opening, leaf area index, and vegetation dynamics (if any)? Anyway, just to present numbers does not help much to improve our scientific understanding.

3.1 temperature The lengthy discussion of global temperature change does not really provide any scientific insight. All it says it that SAI stabilize temperature change and carbon capture maintains 1.5 degree warming by the end of this century, both of which are achieved by experiment design. We can just use a few sentences to cover this info.

Page 9 line 43: What is 'carefully introduced'?

Page 9, line 47: "Because of the careful experiment design here'. Does it imply that previous experiments are not carefully designed? Please rephrase.

Page, 10, lines 87-88 "For example, the mid-century projected PET increase is 102.2 mm/year, but the Sulfur Injection can lower that by 127.8 mm/year, which drops the

absolute value of PET by 25.6 mm/year"

102.2 mm/year is lowed by 127.8 mm/year? Please check the math and expression here.

Regional change: How does the presented PET change for GLENS compare with soil moisture change presented by Cheng et al. (2019)? This should be discussed.

Page 15, lines 50-53: As clearly stated here, there is no direct comparison between GLENS and carbon capture experiment here. First of all, temperature change is not the same, which masks the usefulness of this study.

5. Normalized change To what extend PET change scales with global mean temperature change at regional scale?

Page 16, lines 71-75: There are many studies on the different precipitation sensitivity to CO2 and aerosol forcing. The authors should discuss some of them, in addition to their own study (Lin et al., 2016)

Page 16, lines 76-77: I don't understand how this conclusion is drawn. In this paragraph, only CESM model is mentioned.

Page 16, lines 83-84: I just don't understand this sentence.

Page 17, lines 9-10: Does the contribution from different climate variables to PET add linearly to their combined effect?

Page 17, line 18: What does the weaker sensitivity of sulphur injection mean? It should be 'the sensitivity of XX to sulphur injection is weaker'.

Page 19, lines 56-60: "Instead, we aim to highlight the physical mechanisms at play, especially when distinct between the two approaches (e.g., radiative balances and dynamic response)." I really don't see what insightful physical mechanisms are highlighted in this study.

---

## Author Comment (AC1) · 1 Jun 2020

Reviewer #1

The author studied the climate response to carbon capture and stratospheric sulfur injection geoengineering, using simulation results from two coupled global climate models. The author focused on the climate response, including temperature, precipitation, and aridity over major land area especially North and Southern America.

While the scientific question examined here could further our understandings of the potential impact of different geoengineering strategies on the regional climate response, this paper fails to convince me to be useful in a number of aspects (Discussed below).

Therefore, I don't think the current paper is suitable for publishing unless a substantially improved version is made.

**Response:**

**Thanks for constructive feedback on making the paper more useful to the research community. We have made substantial improvements in this revision.**

First of all, the way the paper is written could be improved substantially. Many important details that are crucial for readers to understand their results are missing, which confuse me a lot during my reading of the paper. For example, in the Methods section, the model and experiment descriptions are just too simple.

**Response:**
**Points are well taken. We have expanded the model description in the Method section (section 2.1). Even though these two models share a lot of common modules, we now provide more details about them (line 115).**

The author compares simulation results from two coupled global climate models, CESM-WACCM and CESM-CAM5, but how different these two models are, and how these differences could affect simulated climate response is rarely included.

**Response:**
**In the previous version, we had mentioned the main difference between these two model versions is the stratospheric component. CESM1-WACCM has a much higher model top and more detailed stratospheric aerosol chemistry and thus is well suited for stratospheric injection simulation. CESM1-CAM5, on the other hand, is the workhorse version of the CMIP5 model and has been used widely (Kay et al., 2017).**

**We now explained that the model difference can affect the simulated climate response because of the climate sensitivity difference, and we have fully acknowledged this caveat and made great attempts to address it by normalizing the response with respect to the temperature response and emphasizing the underlying physical mechanisms instead of the absolute value of responses.**

Also, the author mentioned that the land component differs from each other for these two models, but did not include any discussion about what the difference is.

**Response:**
**The land components are also different between these two versions due to incremental model development. CCR experiment uses CLM4 and GLENS uses CLM4.5. But we do not believe that affects the analysis to a great extent.**
**The difference between these two models is mainly related to climate sensitivity difference that arises from the cloud parameter tuning during the model development process.**

The description of the experiment design and post-processes is also not satisfied. For experiments used in this study, did you use the GLENS project results for the Sulfur Injection case?

**Response:**
**Yes. We did not spell out GLENS in the submitted version but cited the BAMS article promoting it (Tilmes et al., 2018).**
**This is now corrected.**

Where did you get the Carbon Capture simulation results and what is the trajectory of the CO2 concentration/emission in this simulation?

**Response:**
**The carbon capture simulation was done by the first author (Xu) in an earlier study (Sanderson et al., 2017). The trajectory of CO2 emission was explained in Section 2.2b.**

**In this revision, we plotted the emission cut trajectory and the corresponding radiative forcing drops. The absolute values of CO2 emission and concentration are plotted in Sanderson et al., (2017) so we did not repeat it here.**

[Figure]

**Fig 1a of Sanderson et al., (2017).**

For the bias correction, I am confused by how exactly the bias correction is applied to the future simulation cases to account for different climate responses for these two models? etc.

**Response:**
**Bias correction is explained in detail in Section 2.5.**
**Since the bias of present-day simulation is different for these two models, our bias correction scheme will account for that and apply *different* correction factors to the future projection made by these two different models.**

Another example is that in section 2.2, the authors mentioned "CO2 capture and massive CO2 emissions cuts applied to start in 2015-2020", in the next section the author says "both schemes are designed to be deployed in 2020", I am confused which one is accurate? The author should at least state such basic information clearly in the paper.

**Response:**
**Sorry for the inaccuracy. The former is correct.**
**We have now corrected the latter in Section 3.1 to be "Even though carbon capture scheme is introduced slightly earlier in 2015 to 2020, the longer lifetime of CO2 makes the impact of perturbation to the carbon cycle and atmospheric concentration slower to emerge."**

I am also not convinced by the paper. In many places, the author compared the difference of climate responses to two geoengineering scenarios. However, these scenarios are simulated using two different global climate models, and the authors have mentioned that climate sensitivity differs substantially for these two models. How should I believe the difference in climate responses is due to different impacts on these geoengineering schemes, not just because of the difference in the structure of these models? Section 5 is helpful in some way, but I am still not sure how should I interpret most of these results?

**Response:**
**This is the major comment and a deep question.**

**The different impacts of these two geoengineering schemes are what we set out to quantify. Therefore, we must address the limitation of the current experiment set up – two large ensembles are from two related but different climate models.**
**As we now increasingly emphasized in this revision, we highlight four approaches to minimize this limitation:**
**(a) bias correction (see a more technical response below),**
**(b) normalization (Section 5 as the reviewer acknowledged),**
**(c) interpretation of physical mechanism especially the role of solar dimming at the ground surface which is only strongly operating in the sulfate injection case (Section 4)**
**(d) further corroboration of the physical mechanisms at play, using other previously published simulations including volcanic eruption and tropospheric aerosols (Section 3).**

**After all, we call for more coordinated experiments to systematically examine various geoengineering schemes (including carbon capture), which is currently missing in the literature.**

Many descriptions presented in the paper seem not accurate to me.

**Response:**
**Most of the minor comments below involve inaccurate statements in the original
submission. We now made clarification as recommended by the reviewers.
Thank you!**

For example, the authors claim that "spatial patterns show similar agreement between P/PET and
soil moisture as well as P-ET" in figure 2. While the time series do show similarities between
different indexes, the spatial patterns show opposite responses in many regions, especially
Eastern North America and Southern South America. I am not sure if this could be seen as
similar?

**Response:**
**Yes. The purpose of Fig 1 and 2 is to justify the use of P/PET as the main metric since we
do not want to present an analysis that's too complex using multiple drought metrics.
The stronger similarity in time series in Fig 1 is expected because the numbers are the
average over a relatively larger region.
In contrast, the detailed map presented in Fig 2 exposes fine details of these metrics and
their subtle differences over smaller regions, as expected (e.g. Eastern North America and
Southern South America as the reviewer correctly pointed out).
However, we stress the overall similarity between responses of P/PET, soil water, and P-ET
at continental scale including western US and Amazon regions (boxes in Fig 5a) where the
drying trend is largest and passing the significant test.**

Line 7, the authors claim that "it is essential to apply the bias correction to model output before
carrying out a meaningful comparison of future changes". If you are using simulation results
from the CESM_RCP8.5 2010-2019 period to correct both WACCM_RCP8.5 and
WACCM_SulfurInjection, and since WACCM_SulfurInjection is the same to WACCM_RCP8.5
before 2020, why should the difference between these two cases change compared to results
without bias correction?

**Response:**
**The difference between WACCM_RCP8.5 and WACCM_SulfurInjection will yield the
"impact" of sulfate injection. The result will change compared to raw data without bias
correction because, for precipitation and PET, the bias correction is done via
multiplication/division, not simple addition/subtraction.
In the case of T itself, the reviewer is correct that the bias correction applied to future
simulation will not change the difference between the pair and will yield identical results
regard to the impact of sulfur injection.
We now make the technical note above in section 3.1.**

Can you compare these simulations using the same model for all these simulations? Otherwise, I
am not sure what is the best way to account for difference caused by using different models.

**Response:**
**That will be the dream simulation we are proposing in the end!**
**Please see the response to the major comment above.**

Line 10-12: First of all, I think the author should emphasize more that the analysis here is only for changes over land, especially when you are presenting the results.

**Response:**
**Yes. Although we spelled out "major land region" in the figure legend, we did not explicitly state it in the text. This is now corrected.**

The author claims that the larger warming over land for WACCM is due to the larger climate sensitivity of the model. The two models differ not only in climate sensitivity but also the ratio of the land to ocean temperature responses. How much of this larger warming is due to larger climate sensitivity, and how much is due to different land/ocean temperature responses?

**Response:**
**This is a good point and we checked as suggested. The land/ocean ratio for WACCM and CESM are both ~1.5, thus we continue to emphasize the main discrepancy coming from climate sensitivity due to model structural differences, and we continue to try to mitigate this caveat by focusing the discussion on the relative values and physical mechanism, as opposed to absolute values which are model-dependent anyway and thus less useful when broadening this kind of analysis to multiple models.**

The same concern applies to the next paragraph (lines 16 to 20). How much of the larger warming in WACCM compared to other models is due to climate sensitivity or land/ocean responses in these models? If the main reason is climate sensitivity, then the larger cooling in WACCM could just because WACCM produces more warming and the goal is to stabilize global mean temperature changes at the same level. I don't think the reason is simply that a larger injection amount or the injection strategy. If the main reason is the different land responses, then does the explanation in the previous paragraph using climate sensitivity still hold?

**Response:**
**It's hard for us to trace down other sulfur injection model experiments to verify the land-ocean warming ratio, so we have removed the statement of WACCM cooling being larger than others. But we do note that the sulfate injection here is introduced off tropics so that a larger cooling is expected while keeping the injection amount the same. The revised paragraph reads:**
**"The Sulfur Injection simulation here leads to a cooling of 6ºC towards the end of the century, compared with the baseline warming. This larger cooling is designed to largely balance the projected warming by introducing a large amount of sulfur gas. Due to the experimental design that the injection location is off the tropics, this model would produce a greater temperature response than the injection solely from the tropics using the same model (Tilmes et al., 2017; and Kravitz et al., 2018)."**

Also, I don't know what you mean by saying "the longer lifetime of CO2 makes the impact of perturbation to the carbon cycle and atmospheric concentration slower to emerge". Factors that affect the cooling effect of CO2 capture include the inertia of the ocean, the response of the global carbon cycle, etc. I don't know what does the longer Co2 lifetime means here.

**Response:**
**This is exactly the type of physical mechanism distinction we want to highlight between two types of geoengineering approaches. So thanks for asking.**
**It's well known that compared to the duration of aerosols floating in the stratosphere from months to years, the lifetime for CO2 is about decades to centuries. What that means to the radiative forcing perturbation (which is proportional to CO2 atmospheric concentration) is that its response to the emission change (or capture) is much slower than aerosol's response. This is illustrated in Figure 4 (e and f).**
**The inertia of the ocean applies to both cases, and that's why we said "*additional* inertia in CO2 mitigation…)**

The last sentence point to Table 2, column 3. I don't know where to find these results?

**Response:**
**Sorry. This statement refers to an earlier table in which global temperature and land temperature are reported separately. Now we have deleted this statement.**

Also, the author saying "if only the land variables are considered". Do you have any explanation for this?

**Response:**
**The cooling as a fraction of the baseline warming is larger over land compared with the global average, is due to the land-ocean warming contrast ratio as we discussed above. This is not particularly sensitive to forcing introduced (carbon capture or sulfur injection), and thus we chose not to continue to emphasize it.**

Line 54 to 56: From the time series plot, you showed a decrease in precipitation by comparing results between WACCM_SulfurInjection (2086-2095) - WACCM_SulfurInjection (2010-2019), but in table 2, the decrease in precipitation for end of century is 121.0 mm/day compared to WACCM_RCP8.5, smaller than the increase in WACCM_RCP8.5 compared to present day (123.2 mm/day) value. If I understanding this correctly, the present-day value should be the same for WACCM_RCP8.5 and WACCM_SulfurInjection because they branched from each other in 2020. So why is there a decrease instead of a slight increase in precipitation (am I reading this wrong here)?

**Response:**
**Thanks for catching the details. Yes. We meant that looking at the blue line in Fig 3d, WACCM_SulfurInjection precipitation is trending down (slightly) from 2020 to 2095. Comparing 2086-2095 with 2010-2019, the reviewer is correct in pointing out the WACCM_SulfurInjection will actually see a negligible increase of 2.2 (123.2-121.0)**

**mm/year, mainly due to the increase of about 10 mm/year during the ten years prior to the injection starting at 2020.**
**The quantitative statement is now revised at the end of Section 3.2, without impacting the argument.**

Line 102-104: It is clear from Fig.4d that the P/PET index has different changing rates before and after 2050 and since changes in PET is quite consistent throughout the simulation, the difference is mainly caused by different responses in P (Maybe you should do the same calculation and see how much these two terms contribute to the difference in P/PET).

**Response:**
**Yes. We have now calculated the changing rate of P and PET separately to aid the interpretation of P/PET changes.**
**Please see the revised Fig 4 and the numbers quoted in Section 3.3 (the last paragraph).**

However, the authors attribute these differences in P simply to "when the surface cooling starts to emerge, the precipitation also weakens in response to Sulfur Injection". What do you mean by saying "the surface cooling starts to emerge"? The changes in P/PET results seem interesting to me, but I don't think the authors explain these changes in an appropriate way here. I don't even understand what you mean by saying "when the surface cooling starts to emerge" here?

**Response:**
**This refers to the statement at the end of Section 3.3, discussing the response of P/PET in the near term vs. long-term (before and after 2050), and how it differs from Sulfur Injection and Carbon Capture. The phrase "when the surface cooling starts to emerge" is too vague (sorry for the confusion), so we have written the paragraphs completely. It's partially copied here for easy references:**
**"This non-monotonic behavior in Sulfur Injection induced a P/PET response that is highly distinct from the Carbon Capture case in terms of timing. The latter, in contrast, always falls behind the Sulfur Injection changes in inducing climate responses (green curve vs. blue in Figure 4a-d). The P/PET enhancement due to Carbon Capture (compared relative to baseline warming, not relative to present-day), only starts to be significant toward the end of the century with a growth rate of 1.4%/decades after 2050, almost three times larger than the decades before 2050."**

Section 5 investigated the contribution of P and PET to the change in P/PET for different scenarios. For the discussion of P, you compared the precipitation results of stratospheric sulfur injection with tropospheric SO2 increase, and then say "the larger slope of precipitation for Sulfur Injection is expected." Could you directly compare these two results under different forcing designs? I am not sure how could you then get the conclusion that "the model dependence is the largest uncertainty" when you are not comparing the same kind of forcings?

**Response:**
**Thanks for catching that. The discussions related to the synthetic analysis combining the current data with our previously published results (using 20th-century simulation).**

**We have now removed the statement on "the model dependence is the largest uncertainty" because of a lack of justification.**

Also, for the regression results, did you check the standard deviation for these regression results? How robust the regression results are?

**Response:**
**It's unclear whether the reviewer is referring to Table 4 or Figure 8. In Figure 8, the uncertainty range of regression lines is reported in the figure header, with the PET regression onto T being the tightest one. In Table 4, similar statistics had been shown in Lin et al., (2016) so we did not repeat them here.**

It is interesting that global major land PET scales almost linearly with global mean T. If the PET is such tightly related to the T change, why is there a difference in PET slope between these two forcings? I don't think the authors explain it in a reasonable way. If you attribute this also to model difference, then maybe you should use the same model to do the comparison.

**Response:**
**This seems to refer to Fig 8b. Yes, you are right, global major land PET scaled linearly with global mean T. But still, different forcing (SO2 or CO2) can leads to different PET sensitivity. Lin et al. 2015, JGR has shown using the same model, GHG and anthropogenic aerosols, CO2, black carbon, SO2, can all impact PET through different mechanisms, notably the role of surface available radiation. See below.**

[Figure]

**Figure 3.** The effects (in percent change of PET per degree) of relative humidity (RH), wind speed ($u_2$), surface air temperature (SAT), and available energy (Rn-G) on the percentage changes in potential evapotranspiration (PET) over global land due to black carbon aerosol, sulfate aerosol, and $CO_2$, scaled by global mean surface air temperature change. The results of greenhouse gases (GHGs) and aerosols from L. Lin et al. (submitted manuscript, 2015) are shown for comparison. The error bar denotes two standard deviation.

**This decomposition of PET changes is now repeated in Table 5 and Figure 9 to better explain the different slopes of PET in a reasonable way.**

---

## Author Comment (AC2) · 1 Jun 2020

**Referee #2**

Xu et al. compare the changes in aridity under a worst-case warming scenario (RCP8.5), an extreme mitigation and carbon dioxide removal scenario (that I'll refer to as CDR) and the GLENS sulphate aerosol injection scenario (that I'll refer to as GLENS). They combine results from 2 versions of the same base model (with importantly different climatologies) CESM1, which ran the CDR scenario, and WACCM, which ran the GLENS scenario, to assess the surface hydrological response to these different scenarios. To make the two ensembles comparable bias-correction is applied such that both models are adjusted to match the observed climatology.

The authors evaluate a range of hydrological measures and settle on P/PET as their choice of aridity measure. Focusing on this metric they evaluate changes over the global land area (I'm guessing this is what is referred to as "major land") and across the Americas, and evaluate the differences between the CDR and GLENS response. They also normalize the response by global temperature to address a mismatch between the CDR and GLENS experimental objectives.

They find that RCP8.5 produces a general aridification over the land and over the sub-regions they focus upon, driven by a large increase in PET with regionally divergent P responses. They find that while CDR generally partially offsets these trends, GLENS goes too far producing a substantial net reduction in PET that more than offsets the reductions in precipitation. The Amazon is an exception for GLENS, where instead of producing a net increase in P/PET instead it only partially offsets the substantial reduction seen in RCP8.5.

**Response:**
**Thanks for the excellent summary which says it all.**

General comments

My main reservation about this paper is that I don't believe comparing these scenarios is policy-relevant or that scientifically useful. The title sets this up as a competition (vs.) between two geoengineering options but these policies are complementary.

**Response:**
**This is a good point. It's not our intention to set up this comparative study as a competition and selection practice. We have changed the title to reflect this " a comparison between Carbon Capture and Sulfur Injection"**

The CDR scenario (carbon capture as it's often referred to) is also not primarily a CDR scenario but largely an emissions reduction scenario as compared to RCP 8.5 with net negative emissions only after 2050.

**Response:**
**Even though a net negative emission is achieved after 2050, the CDR capacity is assumed to scale up between 2020 and 2050, gradually taking over the contribution by emission cut. For the purpose of comparing the climate response, we did not aim to differentiate the extract contribution of both.**

**We now acknowledge the definition in Section 2.2.b** *(We show the mass amount of CO2 reduced (either through emission cut or various ways of carbon capture) and the associated negative radiative forcing in later figures to illustrate the impact of these idealized trajectories."*

The other issue is that the experiments were designed to achieve different objectives and produce quite different radiative forcing and temperature responses over the analyzed period and were run in model-versions that differ in quite important ways. Both of these factors undermine the utility of this comparison.

**Response:**
**We acknowledge that the two model versions are not exactly the same, which may hinder some analysis. We had made attempts to mitigate that as detailed in later responses. It is our ultimate goal to inspire more coordinated model experiments (from multiple groups hopefully) in the future, which is currently missing.**

**Despite the lack of coordination when these two sets of the simulation were produced a few years ago, we do not agree that these two experiments were designed to achieve fundamentally different objectives. They both aimed to achieve a temperature stabilization towards the end of the 21st century (at slightly different levels per Fig 3c), and this long-term equilibrium provides a great opportunity to study climate response.**

That said, clearly a lot of careful analysis has been done and so I'll continue to my other comments.

**Response:**
**Thanks for acknowledging the comprehensiveness of our analysis.**

Much of the paper is written as if the two experiments had the same aim but they didn't. The methodology doesn't make the different goals of the two experiments clear enough and their high-level difference ought to be described up front, i.e. section 2.2 should be elaborated upon to make clearer the experimental setups and their differences.

**Response:**
**The suggestion of making the experimental objectives clear is well taken. We now acknowledge that the two experiments were designed and conducted by different groups upfront in the new opening paragraph of Section 2.2.**
*Two pairs of model simulations were conducted and published in the last few years, both featuring a large ensemble to enhance the robustness of examined climate responses, especially at a regional level. The WACCM set (http://www.cesm.ucar.edu/projects/community-projects/GLENS/) is aimed at stabilizing the climate at its 2020 level with sulfur injection. The CESM set (low warming large ensemble, http://www.cesm.ucar.edu/experiments/1.5-2.0-targets.html) is aimed at climate stabilization at 1.5 and 2ºC warming levels, with aggressive emission cut and rapid growth of negative emission technology.*

There is often reference to "lagging effects" of the CDR experiment but given that it had different ends from those of the GLENS experiments it seems inappropriate to describe the temporal evolution as if it were trying to achieve the same thing.

**Response:**
**The "lagging effects" of CDR is a point we want to emphasize so thanks for bringing it up. As shown in Fig 4 (the temporal evolution of perturbation and response), the CDR measures actually lead the Sulfur Injection (Fig 4e), so there is indeed a lagging in the achieved forcing (Fig 4f). Note that the temperature response (Fig 4a) to the forcing will also lag (due to ocean thermal inertia), but irrelevant to the forcing imposed.**

I was surprised that transpiration wasn't discussed as this is a major driver of terrestrial hydrology. I strongly recommend considering the direct CO2 physiological effect of CO2 in these simulations as this will be a big difference between CDR and RCP8.5 / GLENS.

**Response:**
**Good point, and we add a note in the Conclusion section.**
*The direct physiological role of CO2 is potentially important because the CO2 level is greatly reduced in only one of the two mitigation approaches. But this study, focusing on meteorological drivers of land aridity using P/PET, did not delve into CO2's suppression on plant transpiration via stomatal closure, which also appears to be weak in these two models compared with other climate models (Swann et al., 2018)*
**The weaker physiological effect is in response to comment later.**

**As for the calculation of PET, we formulate it as a function of temperature, relative humidity, available energy, wind, and bulk stomatal resistance (rs). We indeed employed a fixed bulk stomatal resistance (i.e., 70 s/m) while those values can vary over different regions and time as the reviewer suggested.**
**The simplification is justified by that even adopting widely different but reasonable choices of vegetation parameters including rs, the PET values do not change much (within 5%). Scheff and Frierson (2014), Fu and Feng (2014) and Fu et al., (2016) showed:**

The change of PET by neglecting the higher-order terms can be written in the form

$$\Delta PET \approx \frac{\partial PET}{\partial SAT} \Delta SAT + \frac{\partial PET}{\partial RH} \Delta RH + \frac{\partial PET}{\partial (Rn - G)} \Delta (Rn - G) + \frac{\partial PET}{\partial u2} \Delta u2 + \frac{\partial PET}{\partial r_s} \Delta r_s \quad (1)$$

$$\frac{1}{PET} \frac{\partial PET}{\partial r_s} = -\frac{\gamma C_H |u2|}{[\Delta(SAT) + \gamma(1 + r_s C_H |u2|)]}, \quad (2)$$

**Note that the magnitude of the PET relative change is largest when rs=0, then it goes down to be less than -0.4%/ (s/m). The negative sign means that, with elevated CO2 level, a larger stomatal resistance leads to smaller PET (implying a terrestrial wetting if everything else is fixed).**

The authors report results showing PET and ET over land, finding that they both increase in lockstep. This was surprising to me, given the results of Swann et al. (2018) who found that PET was projected to increase in models while ET was not. Swann et al. found that this was due to reductions in transpiration due to CO2 fertilization offsetting the meteorological drivers that drove PET increases. This suggests that the CO2 physiological effect has not been included or is very weak. Please provide details and analysis to address this.

**Response:**
**This refers to Fig 1(a). ET increase in WACCM simulation but note that ET increase is 100 mm/year less than increase in PET). The ET increase is also consistent with decrease in soil water in top 10 cm of soil.**
**But, Why an increase in ET? Cheng et al. (2019), co-author of the present study, has looked into that carefully and found that global mean ET is indeed increasing including the canopy transpiration part under RCP8.5. We also checked distribution of canopy transpiration (Bottom row in the next figure), with increase in most land area under RCP8.5, except Southeast Brazil and Southern Africa.**

[Figure]

**As speculated by the reviewer, in these two models,** increases in soil liquid water and canopy intercepted water under RC8.5 (red lines in the next figure) overwhelms the direct physiological effect of $CO_2$ (suppressing transpiration) in RCP8.5.

[Figure]

**The different model behavior is also one reason that we did not delve deep into analyzing other direct land model output.**

Relatedly, it is not clear whether there were any differences in land cover between the CDR scenario and the RCP8.5 / GLENS scenarios. This could have a large effect on regional hydrology.

**Response:**
**To our knowledge, land cover and land use are not perturbed in CDR or GLENS scenarios and thus should be consistently the same as in RCP8.5. We add this note to Section 2.2.**

The temperature difference between the two experiments should be made clear, from table 2 I can back out that CESM-1.5C is 0.7C / 0.5C warmer than the baseline whereas GLENS is 0.1C warmer (is this right? I thought it was designed to perfectly offset warming from the baseline period).

**Response:**
**Those are correct. GLENS is designed to be stabilized at the 2020 climate, while our baseline is 2010-2019 and thus the 0.1C difference.**

Given the lower sensitivity of CESM, this 0.5C figure would be larger if both were run in WACCM. This could be driving some of the difference between these two experiments and should be brought out and discussed.

**Response:**
**Not necessarily. The 0.5ºC is the experiment design that aims to stabilize the climate at 1.5ºC inspired by the Paris Agreement. (the 2015 warming is 1ºC relative to pre-industrial) Had the WACCM been used for this simulation, we would change the emission trajectory to meet the 1.5C level as well so to maintain the 0.5ºC difference.**

I'd like a clearer sense of the magnitude of the bias-correction overall and of the regional character of it in your study region. The fact that WACCM has around 25% more precipitation over "major land" (undefined as far as I can tell) ought to be highlighted!

**Response:**
**Let us clarify.**

**The magnitude of the bias-correction overall can be seen from Fig 3 (first two rows). The magnitude of the bias-correction of the focused regions can be seen by contrasting Fig 2 (uncorrected) and Fig 5 (corrected).**
**From Fig 2b, WACCM has around 5% more precipitation than CESM, not 25%.**
**Te "major land" was defined in Table 2: "the land regions over 60ºS to 60ºN, thus excluding cold regions where the seasonal snow cover or permanent ice sheet surface makes P/PET less useful as a predictor for aridity and vegetation types."**

Beyond this the regional biases ought to be made clearer. Are the models far off in key regions such as amazon? How wrong are the arid and semi-arid areas calculated using the models uncorrected P/PET values? How different are the regional biases? An extra figure or two is needed to make this clear.

**Response:**
**Thanks for asking these questions.**
**Comparing the observation in Fig 5a, and the uncorrected models in Fig 2a and Fig 2g, the models appear to be doing a reasonable job in capturing present-day P/PET, over Amazon, and also over the arid/semi-arid transitional regions. Of course, the corrected model in Fig 5b and c have matched with observed P/PET nicely.**
**Given the number of tables and figures already included, we hesitate to include more display items.**

The temperature normalization section doesn't seem to add much to the paper and has some serious problems. I understand that it could be useful for comparing scenarios with different levels of cooling if it were not for the difference in the climate sensitivity. WACCM's climate sensitivity looks to be ~20% higher than CESM's which will mess with this normalization procedure.

**Response:**
**Section 5 on normalizing the regional change relative to global cooling numbers is specifically added to address the caveat that the two models have different climate sensitivity.**
**Our rationale is that normalization can be used to tease out both the forcing differences (as in our previous paper of Wang et al., 2017 Sci Report using a single climate model), as well as model differences.**

I'd suggest either cutting this section (what do we learn that isn't covered elsewhere?) or addressing this climate sensitivity issue by testing how different the RCP8.5 sensitivities are between these 2 models. I worry that model differences rather than scenario differences could be driving some of the response seen in this section.

**Response:**
**Actually as argued by Reviewer #1, the model difference rather than scenario difference will affect the absolute values of the response reported in previous sections to a greater extent than in this normalization section. Therefore, it is our intent to keep Section 5 as a**

**mitigation to the model difference issue. We also elaborate on this rationale in the Conclusion section.**

**"An important note should be made regarding the interpretation of the quantitative "benefits" presented. Because these two models have different climate sensitivities, the baseline warming induced by unchecked emissions growth is not at the same level. The WACCM warms faster, reaches the 2ºC level at earlier decades, and has higher end-of-century warming at 6ºC, compared to the CESM. Since the baseline warming is different for the two models at different decades, the climate "benefit" due to any mitigation measures shall also be interpreted in a relative sense, i.e., the fraction (%) of the projected change in the future that can be mitigated by the Carbon Capture or Sulfur Injection. For example, even if our results suggest Sulfur Injection can lead to 6ºC cooling while the Carbon Capture can lead to a 4ºC cooling, that does not quantitatively provide any constraints onto the strength of respective approaches. "**

The section summaries seem unnecessary.

**Response:**
**Because of the lengthy mixture of results and discussion in Section 3 and 4, we still keep the section summary for clarifying the main points.**

I think it would be more fair to describe the "carbon capture and storage" scenario as an extreme mitigation scenario. A reduction in positive emissions makes up the bulk of the difference between RCP8.5 and this scenario.

**Response:**
**We agree. We now clearly explained in Section 2.2 that the CESM experiment is aiming to test the impact of an "extreme mitigation scenario" that contains a mixture of aggressive decarbonization and carbon removal.**
**"The CESM set (low warming large ensemble, http://www.cesm.ucar.edu/experiments/1.5-2.0-targets.html) is aimed at climate stabilization at 1.5 and 2ºC warming levels, with aggressive emission cut and rapid growth of negative emission technology, an extreme mitigation scenario."**
**We also note that for the physical climate system to respond, one tonne less emission would be the same as one-tonne carbon captured.**

If you wish to highlight the use of carbon dioxide removal (CDR) or the presence of negative emissions then I'd suggest using these terms instead of carbon capture and storage as this technology can be used without producing negative emissions, i.e. on coal power plants.

**Response:**
**The negative emission technology has too many different terminologies already these days, carbon and storage, carbon capture and sequestration, carbon dioxide removal, direct air capture. We do want to avoid getting into further complexity and thus retained the use of "carbon capture" as a short-hand.**

**But, in Section 2.2b, we thoroughly clarify what we really meant by "carbon capture".**

*"with CO2 removal and massive CO2 emissions cuts applied to start in 2015-2020 (hereafter referred to simply as "Carbon Capture")*
*"We show the mass amount of CO2 reduced (either through emission cut or various ways of net emission technology) and the associated negative radiative forcing in later figures to illustrate the impact of these idealized trajectories. However, the technical feasibility and required socio-economic shift to facilitate the scale-up of capacities such as direct air capture (Hanna et al., 2020) and clean energy transition (Hanna et al., 2020b) is beyond the discussion of this paper."*

**Also in the abstract, we now clearly state: "The CO2-based mitigation simulation is designed to include both emissions cut and carbon capture."**

I'd suggest coming up with some clear shorthand for the experiments and using it consistently in both text and figures, e.g. CESM-RCP8.5, CESM-CDR, WACCM-RCP8.5, WACCMGLENS, Baseline.

**Response:**
**Yes. The shorthand is now consistently used as CESM-RCP8.5, WACCM-RCP8.5, CESM-CarbonCapture, WACCM-SulfurInjection, CESM_historial.**

The figures have inconsistent labeling, line colours and styles and some of the captions are oddly formatted. The manuscript text needs a careful proof-read by a native English speaker. There were too many grammar mistakes so I only addressed the worst. There were also many very short paragraphs that could be merged with their neighbors.

**Response:**
**The figures and text will be thoroughly revised in the next version. Short paragraphs were merged where appropriate. Thanks for the careful comments below.**

Specific Comments

N.B. Specific comments are given in the order that they appear in the manuscript with the line numbers as I saw them. It seems the pdf has cut off the hundreds part of the line number and I haven't converted the cycling line numbers into something more sensible.

**Response:**
**Yes. We use the cycling line numbers in the published pdf.**

Title – Given that you stress in the conclusion that you aren't trying to evaluate which is better and instead are focused on the mechanisms, I'd suggest: "climate engineering and aridity in the Americas: a comparison of carbon dioxide removal and sulphate aerosol injection"

**Response:**
**We agree and we change the 2nd part of the title to avoid the impression that the two are in competition. It is now "a comparison between Carbon Capture and Sulfur Injection".**
**We stick with carbon capture as explained previously. We did not use SAI (a common shorthand though) because technically what's injected is gas, not aerosol.**

22 – I'd argue that this paper hasn't used a consistent framework: different models and different objectives for deployment

**Response:**
**We now tune down the language a bit.**
**It now reads: "Despite being in the public debate for years, these two leading geoengineering schemes have not been examined under a consistent analytical framework using global climate models. "**

22-24 - Given that this is more a mitigation scenario than a pure-CDR scenario, it's not correct to describe this as the first paper to compare sulfate geoengineering against "carbon capture" as previous studies have made such comparisons: [Niemeier et al. 2013: DOI:10.1002/2013JD020445; Muri et al. 2018; DOI:10.1175/JCLI-D-17-0620.1; Jones et al. 2018, DOI: 10.1002/2017EF000720, etc.].

https://agupubs.onlinelibrary.wiley.com/doi/full/10.1002/2013JD020445
https://journals.ametsoc.org/doi/full/10.1175/JCLI-D-17-0620.1
https://agupubs.onlinelibrary.wiley.com/doi/full/10.1002/2017EF000720

**Response:**
**Thanks for suggesting the references, which are now cited in the conclusion section to provide a broader context.**
**" (2) provide broader insights to the mitigation impact of other geoengineering approaches beyond the two discussed here (such as cirrus ice cloud thinning or marine warm cloud brightening, Muri et al., 2018; and surface albedo modifications, Crook et al., 2015; space mirror, Niemeier et al., 2013). "**

**Of the three papers suggested, only the Jones et al., (2018) is somewhat close to our analysis, in which SRM is indirectly compared with RCP2.6 which has a weak deployment of CDR. Note that CDR in RCP2.6 is much weaker than what is in our CESM experiment (1.5C).**

[Figure]

**Fig 1a of Sanderson et al., (2017).**

**But even Jones et al., (2018) did not explicitly compare SRM against the carbon mitigation scenario as suggested by the reviewer.**

32 – what does "mitigation potential" mean in this context?

**Response:**
**It means the potential mitigation benefits via climate engineering. Since all of these are hypothetical by design, the benefit or impact is now unrealized potential. We change it to potential mitigation benefits in this revision.**

39 – does it worsen the trend or is the trend worsening under RCP8.5?

**Response:**
**The trend is worsening under RCP8.5. The sulfur injection is not very effective in curbing it.**

67 – Vaughan and lenton 2011 don't provide evidence of investment, do they?

**Response:**
**We remove the statement of seed investment in this revision.**

68 – I thought CDR installations would be effectively independent of emission sources. CCS is installed directly onto power plants, etc. but that's classed differently.

**Response:**

**Yes. The CDR installations that can operate effectively independent of emission sources are often referred to as direct air capture (see our recent work: Hanna et al., 2020).**
**CCS is the more "traditional" tech that would operate mostly in a high concentration environment in the power plant (e.g. to be combined with biofuel energy sources, BECCS). No change is made to the text.**

89 – worth comparing that to the "dry-gets-drier" pattern of global warming

**Response:**
**Yes. We now stated the sulfur injection response is opposite to global warming.**
**"Based on the analysis from Simpson et al. (2019), the precipitation in tropical and extratropical regions shows a dry-get-wetter, wet-get-drier pattern due to the aerosol induced stratospheric heating, opposite to the well-known pattern of dry-gets-driver response due to global warming."**

Sect. 2.1 This is a very short model description, and includes only one citation to one of the model versions used. Please elaborate.

**Response:**
**This is now greatly expanded as suggested by Reviewer #1 as well, particularly to note the model difference and impact on the results.**

Sect 2.2 Both experiments need to be described in more detail.

**Response:**
**This is now expanded with an opening paragraph to state the objective of two experiments upfront, as well as acknowledging the "carbon capture" also includes some form of emission cut.**

30 – "to stabilize TEMPERATURE at 2020 levels. . ." would be more accurate.

**Response:**
**Yes. "climate"->" temperature".**

35-41 – Is this based on another RCP, How large are the negative emissions, are there any differences in the land surface (more forest cover, etc.)? This description leaves out crucial details. Please outline them here even though they are explained in detail in the Sanderson paper.

**Response:**
**It's a special scenario constructed by a group of us. We now added "(hereafter referred to simply as "Carbon Capture"), which is constructed to have a more aggressive decarbonization pathway than RCP2.6 (Sanderson et al., 2017)."**
**We also added in Section 2.2 that "Land cover and land use in these two pairs of simulations is consistently the same as in RCP8.5. "**

Sect 2.3 – This is too little material for a sub-section, so I'd suggest cutting it or else expand it to discuss more of the analysis approaches employed, e.g. the time-periods covered and the procedure to normalize by global temperature change that is employed later.

**Response:**
**The suggestion is taken. We now merge it into the opening paragraph of Section 2.2.**

65-59 – This seems out of order, I'd suggest moving figures 1 and 2 out of the methods section and into the results section.

**Response:**
**Although uncommon, we'd like to retain the first mentioning of these two figures in the method section, mainly because they are secondary and confirmatory and do not provide direct results.**

Figure 1 – It would be much clearer to color one axis red and the other blue and use dashed versus normal to separate the experiments, then one could read the figure at a glance rather than having to get half-way through the description to know what is shown.

**Response:**
**Revised as suggested.**

Figure 1 – This shows ET and PET increasing in lock-step but Swann et al. 2016 (www.pnas.org/cgi/doi/10.1073/pnas.1604581113) showed all climate models they investigated diverging with PET rising rapidly and evaporation not rising at all. Swann et al. argued that this was due to the direct physiological effect of CO2 suppressing transpiration. That would suggest that there is no direct physiological effect in your simulations, is that correct? If not, can you explain the difference between your results and those of Swann?

**Response:**
**The physiological effect of CO2 is included through the CESM and WACCM simulation but appears to be weaker than other models and the suppression effects are only strong over tropical forest regions.**
**See the response to general comments as well.**

Figure 2 – "significant" – please elaborate here or in the methods.

**Response:**
**Added. "where the differences are significant at 95% confidence level following the student's T-test"**

L80-84 – These descriptions are incomplete. I cannot tell whether this is a reasonable approach as the terms are not defined.

**Response:**

We now remove the equation due to undefined terms. Instead, we further clarify in the text.

89-99 – This paragraph is hard to follow, I'd suggest revising it.

**Response:**
**We now revised it thoroughly to bring clarity. It's now shortened to be "The same bias correction is applied to future simulations (CESM_RCP8.5 and CESM_CarbonCapture, in 2006 and afterward), which branched from the CESM_Historical in 2005. For WACCM_RCP8.5 simulations, we indirectly adjust them to agree with the corrected CESM_RCP8.5 in 2010-2019 ( "present-day"). We could not adjust WACCM directly based on observations because the WACCM-RCP8.5 simulation only starts in 2010 and overlaps with the observation record (ending in 2019) for less than ten years. Similarly, WACCM_SulfurInjection was adjusted to match with WACCM_RCP8.5 in 2020 (when it branches from WACCM_RCP8.5). "**

89-91 – Is the same bias correction applied to both models or are the separate historical bias corrections applied to the future in both models.

**Response:**
**Bias correction is separately done to these two models because they apparently have a different historical bias (Fig 3 a and b).**

93-94 – I don't believe this is the same "present-day" as in the GLENS experiments which I believe aim to keep conditions fixed at their 2010-2030 levels.

**Response:**
**No. "present-day" is here defined to be earlier (prior to 2020) so that there is no difference between these simulations at present-day (Fig 2c).**

Section 3.1 – The sub-section title seems to be at odds with the section title, perhaps change the section title?

**Response:**
**You are right. We change the section title to be "Mitigation at the global scale".**

Figure 3 – What is "major land"?

**Response:**
**See the responses earlier and Table 2 caption.**

`Fgiure 3 – The change in axis range between b and d should be avoided. You should make clear visually or in the text that WACCM's precipitation over "major_land" is ~25% too high. This is huge!`

**Response:**

**Different models have different climatology. That is why the community showed anomalies for CMIP5, not raw output. We now make a note about the P bias in both models. "For example, comparing Fig 3b and d, WACCM appears to have a 25% positive bias for precipitation over land, and CESM has a smaller positive bias."**

10-11 – What is WACCM's climate sensitivity? If it's not known then I'd reverse the order and highlight that it's higher than the already-high 4 C of CESM.

**Response:**
**We added WACCM climate sensitivity now.**

13-14 – This is the first time that this has been mentioned, I'd suggest highlighting this fact in the methods section.

**Response:**
**Yes. We now mention the climate sensitivity of both models in the method section (the end of Section 2.1).**

16-20 – The off-tropics injection is irrelevant as if the sulphates were less effective more would have been injected. Of course there's also the afore-mentioned high climate sensitivity, which should be mentioned.

**Response:**
**Good point. Since the goal is to stabilize the warming in 2020, the cooling can be enhanced with more injection to achieve that. The paragraph is rewritten as :**
**"The two types of mitigation efforts, by design, would lead to a similar amount of temperature stabilization to a level close to the present-day. The Sulfur Injection simulation here leads to a cooling of 6ºC towards the end of the century, compared with the baseline warming. This larger cooling is designed to largely balance the projected warming by introducing a large amount of sulfur gas, some from locations off the tropics (Tilmes et al., 2017; and Kravitz et al., 2018)."**

22 - "global major land" needs to be defined.

**Response:**
**The definition of "major land" is the land regions over 60ºS to 60ºN. See Table 2 caption for rationale.**

25-28 – Isn't it better to describe this as a fundamentally different experiment given their different aims? The carbon capture and GLENS experiments have different ends that end up producing a roughly similar temperature response, i.e. little change from 2020.

**Response:**
**With respect, we disagree they are fundamentally different experiments because both are designed to produce a stabilized climate that's not too far from 2020, with different means though.**

**Please see the new opening paragraph of Section 2.2.**

Table 2 – Seems unnecessary, suggest cutting. You've already shown this in figure 3, and reported many (perhaps too many) of these figures in the text.

**Response:**
**We want to keep Table 2 in order to support the quantitative statements in the text. Without Table 2, it's hard for us and the readers to keep track of the numerical figures reported in the text.**
**Without Table 2, it would also be hard for reviewer #2 to back out the temperature changes between now and the end of the 21st century.**

29-31 – These lines are unclear, larger than what? Also unclear what is being referred to. Is column3, mid-century P or is it all of the PET results? Do the variables count as a column?

**Response:**
**Sorry for the confusion. There was no Column 3 in Table 2. We were mistakenly referring to an earlier version of Table 2 in which global results and major land results were presented separately. We have removed the statement in question.**

35 – I'd avoid the term "mitigation" given its use to refer to emissions cuts in the climate literature. I'd suggest renaming this "mitigation potential" to something else.

**Response:**
**I think climate engineering, in addition to emission cut, is also part of the mitigation/solution package.**
**We change "mitigation potential" to "potential mitigation benefits".**

37 – explain that this is a reduction of a precipitation increase not a 100% reduction in precipitation.

**Response:**
**Thanks, we corrected it.**

38-63 – These paragraphs on the global-mean precipitation response ought to be revised, they are not well written. It is also strange to refer to an almost perfect reversal of RCP8.5 precipitation trends as being over-effective.

**Response:**
**There might be some misunderstanding here. The over-drying is referring to other sulfur injection experiments previously published. For this model version, the over-drying is dampened.**
**The paragraphs are revised to strengthen the argument.**

L63 – Could you state what "this precipitation-centered argument" is or else reframe this.

**Response:**
**It's explained in the sentence immediately following: "precipitation alone does not reflect the full effects of the hydrological cycle on the terrestrial ecosystem.". We also change "this precipitation-centered argument" to be "this precipitation-centered perspective in assessing terrestrial aridity".**

L67 – I'd avoid referring to sulfate injection as a mitigation scheme.

**Response:**
**We change the "mitigation scheme" to "climate engineering schemes".**

L67 – why aspire to do it and not just get on with it?

**Response:**
**We change "we aspire to include " to " we also included".**

70 – does it? I see a drying in all regions in P/PET and a wettening trend in most places except the amazon and central America.

**Response:**
**We change " a close agreement" to " a broad agreement".**

73-75 – this sentence is mangled.

**Response:**
**Thanks for catching that. Rewritten as "Despite an increase in P, the projected PET growth that approximately scaled with T increase will exacerbate future land aridity".**

75-77 – is this in the RCP8.5 experiment?

**Response:**
**Yes. Added.**

77 – "bring forth the benefit of curbing aridity worsening"

**Response:**
**Revised to be "reduce the tendency of worsening aridity".**

89 – I'd leave this type of commentary until the discussion as its tangential to what is being described.

**Response:**
**Commentary language "(presumably shifting the climatic zone to a category beneficial to the cropland)" is now removed.**

95-96 – Better to describe this as changing the sign of the trend as reversing is ambiguous in this context, i.e. it could mean simply offsetting.

**Response:**
**Great point. We now made it clear that the drying trend is flipped to be wetting, instead of being "reverse". It now reads:**
**"Sulfur Injection appears to have this additional "benefit" of flipping the sign of the drying trend as projected in the baseline scenarios"**

98-99 – the shorter response time doesn't explain this. P/PET rises rapidly then stops rising, Why is this?

**Response:**
**The entire "on the timing" paragraph is rewritten, with the Fig 4 replotted. The P/PET growth rate becomes smaller in the second half of the 21st century because the P decreases at a faster rate than PET decrease, which again is due to the short response time nature of Sulfur Injection.**

Figure 4. Again, these experiments are so different in character it seems odd to compare their time evolution. They have a different temporal evolution of forcing (in ways not described or explored in this paper) so its unsurprising that they have different temporal evolutions of climate response.

**Response:**
**Thanks for asking because this is a key figure with the differences bringing new insights. The different temporal evolution of forcing in (f) is the main point we want to address because the carbon capture, despite having an earlier adoption in the effort (e), lags behind the Sulfur Injection.**

Section 3.4. I'd avoid the generic sub-section title "summary" and give something specific to this section.

**Response:**
**We change the generic sub-section title to "3.4 Summary: P/PET as the aridity metric and the faster/stronger benefits due to Sulfur Injection". We will follow the editorial guideline if this turns out to be too specific.**

10-14 – As I mentioned earlier, the results in figures 1 and 2 do not show the same pattern for the different variables so I don't think it fair to say P/PET stands in for all measures of aridity as this text implies.

**Response:**
**It's never our intent to claim P/PET stands in for measures of aridity. We select it to avoid getting into the weeds of too many metrics. We have now made it clear that Fig 1 and 2 did reveal subtle discrepancies especially at a regional level, and thus more metrics should be**

**included if the goal is for a comprehensive drought condition assessment as opposed to the comparison of two high hypothetical scenarios.**
**We added "Note that we select P/PET among other drought indicators to avoid the complexity of the analysis because the main goal of our analysis is to compare highly extreme mitigation scenarios. Fig 2 revealed regional discrepancies that suggest for a more comprehensive drought assessment, more metrics should be included in addition to P/PET."**

18 – where is the $CO_2$ captured results?

**Response:**
**The mass captured/cut is shown in Figure 4e, a key piece of information.**

21 – This type of discussion should be saved for the discussion section.

**Response:**
**Since we do not have a dedicated Discussion section, and the discussion in the Conclusion section is mostly related to the limitation of the current study and implication for future studies. We still keep this quick comment here.**

Section 4, how about "hydrological change in the Americas"?

**Response:**
**Point taken. We change it to "4 Regional responses in the Americas"**

Figure 5. This figure has too much going on and should be split or else the mid-century results cut.

**Response:**
**We now split Fig 5 into Fig 5 and 6, one for CESM and one for WACCM. We do not want to cut mid-century results because the contrast between the near-term and long-term is a key point of ours.**

The WACCM RCP8.5 results are missing which are crucial for interpreting the WACCM GLENS – RCP8.5 anomalies plotted.

**Response:**
**The split enables us to add WACCM_RCP8.5 results to the first column of the new Fig 6.**

I'd suggest replacing the bias-corrected model results with the original model results. It is unsurprising that the bias-corrected results look very similar to the observations. The models simulated present day has a direct bearing on its projections for the future whereas the bias-corrected present day doesn't.

**Response:**

**We had already shown the original model results in Fig 2, so we did not make the changes. We had also discussed the different biases of the two models and implications on projection.**

I'd also suggest making the CDR and GLENS anomalies relative to present-day rather than RCP8.5.

**Response:**
**The anomaly in relative terms (%) is indeed calculated with the present-day as the reference. i.e, (CDR - RCP8.5) / present_day(2010-2019).**

Finally, The caption format shouldn't include paragraph breaks.

**Response:**
**Of course. Just suppose to make a reviewer's life easier since it's a busy figure. Will correct once sent out for the press. Thanks.**

43-45 – which sees this to a lesser extent? Is the global value "major land"?

**Response:**
**Clarified as although to a less extent "for the latter".**
**Change "global value to " Responses over global major land"**

46-48 – this seems redundant

**Response:**
**We shortened and merged it with the previous paragraph.**

Table 3. The formatting / caption could be clearer here. Is CESM historical the same as historical after bias-correction, are they similar?

**Response:**
**Yes. CESM_historial is after bias correction. Dryland area is based on absolute definite, i.e., AI < 0.65. That means we have to correct the bias. otherwise, we cannot obtain the dryland area.**

51 – make clear that you are referring to the global arid area.

**Response:**
**Global added.**

4.2. This whole South America section is poorly written, I'd suggest revisiting it.

**Response:**
**Suggestion well taken. Many revisions are included.**

61-62 – revise this sentence.

**Response:**
**Revised to be " While the changes over North America are largely consistent with the global land change, South America's responses are more complicated."**

65 – I'd suggest using 2 significant figures here. What does the 15-30% refer to?

**Response:**
**We did not go further to two significant figures because of the inherent uncertainty. 15-30% refers to relative change as shown in Figure 5.**

66-68 – Is the amazon really the only region in the Americas to see a decline in precipitation in RCP8.5?

**Response:**
**Yes. Original Figure 6b shows that North Brazil has a significant precipitation decline under RCP8.5. The decare over Easter Us appears to be less significant.**

Figure 6 – Given the differences in the climates of the regions under investigation would it not be better to report results in percentage change terms? There's a similar ~50 mm/day increase in PET in West US and Brazil but presumably they have very different absolute values.

**Response:**
**Good suggestion. But the percentage change terms are already reported in Table 2 and Figure 5. Also, we do need to present the absolute value changes of P and PET to understand the P/PET change. Using relative changes of P and PET to derive P/PET relative change would be misleading.**

73-74 – this isn't a good description of what is in the figures, P is up in all regions in the CDR experiment which is just different from what is going on under RCP8.5 and around 80- 90% of the PET response is offset.

**Response:**
**We are talking about Northern Brazil here which has quite different P responses compared to other regions examined.**
**The revised sentence is "This is achieved by both offsetting the precipitation decrease over Northern Brazil, but more importantly, by offsetting the projected PET increase by 80 to 90 % (Figure 6 b,c).".**

79 – why not open with the big picture for GLENS then address this exceptional response?

**Response:**

**Unclear to us what the reviewer meant by "big picture". We are well into the section of South America which we mainly focused on North Brazil (Amazon).**

83-85 – it's not consistent with Simpson et al if it's the same data, it's just the same thing.

**Response:**
**Wording changed to be "This additional suppression of tropical rainfall due to Sulfur Injection is also previously reported in Simpson et al., (2019),"**

94 – continues? When did it start?

**Response:**
**Revised to be "Unlike Sulfate Injection, Carbon Capture is consistently capable of mitigating the projected drying trend almost entirely over South America (similar to the global and eastern US cases)."**

96 – There's a larger reduction in PET in GLENS, so the differences should be due to the differences in the precipitation response. Should probably note that the precipitation reduction in CESM-RCP8.5 is around twice as large as in WACCM-RCP8.5.

**Response:**
**I think the reviewer misread the sentences as Amazon ("the precipitation reduction in CESM-RCP8.5 is around twice as large as in WACCM-RCP8.5" as shown in Figure 6b), while it's referring to sub-Amazon.**
**No changes are made except minor wording clarifications.**

00-01 – Which regional trend is being referred to here?

**Response:**
**"over the sub-Amazon" added and the paragraph merged with the previous one on carbon capture.**

11-13 – Again, what is being referred to here?

**Response:**
**It's about "sub-Amazon" over South Brazil.**

Section 4.3. Again, I'd avoid using "summary" for a sub-section title.

**Response:**
**We change it to "4.3 Summary: the unique responses over Amazon (North Brazil)", pending editorial approval.**

37 – it's not a further decrease in precipitation if the decrease in precipitation is less than in RCP8.5.

**Response:**
**It is indeed a further decrease in precipitation because the decrease in precipitation is more than in RCP8.5, as shown in the original Figure 6b.**

50 – again, I'd suggest avoiding using mitigation in this way: "mitigation capacity"

**Response:**
**We change it to "mitigation benefits". We do not see a problem calling Sulfur Injection a mitigation approach.**

52 – Presumably WACCM-CDR (if you had run it) would be about 20% less effective than CESM-CDR by this measure as WACCM has a higher climate sensitivity and so the response would be divided by ~6C rather than ~5C.

**Response:**
**We do not entirely agree. If WACCM_CDR were used, the actual response will also be different than what's in CESM-CDR, so it's not straightforward to claim that the normalized changes would be 20% less effective.**

Figure 7. I don't see any value in including this figure. How is the statistical test applied here?

**Response:**
**Figure 7 (now Figure 8) of the normalized change is a key figure to bring up the physical insights and we had retained it. The scaled change of P/PET induced by carbon capture and sulfur injection can tell the efficiency of different forcing.**
**The statistical test applied is the same as in Figures 2 and 5 so we did not repeat the details here.**

56-59 – This is a result of not changing the scales of the plot to reflect the fact that you've divided it by ~5.

**Response:**
**Correct.**

62 – Higher climate sensitivity is the obvious driver of this but isn't mentioned.

**Response:**
**With respect, we do not think higher climate sensitivity is in play here because we have normalized the change with respect without global temperature change. This seems to be related to the confusion as in the general comment. In other words, the climate sensitivity will affect the spread of the points in Figure 8 (blue vs. red), but not much the slope of lines.**

Figure 8. It is not clear what is going on in this plot. Which model is plotted? Are these the bias-corrected results or not? This plot needs to labeled more clearly.

**Response:**
**As labeled and explained in the caption, Red for carbon capture (using CESM) and blue
for sulfur injection (using WACCM), both bias-corrected.**

I'd suggest producing a similar plot for the RCP8.5 results of the 2 model versions as I suspect
that they could look quite different and this might be driving some of the difference attributed to
the forcing differences here.

**Response:**

[Figure]

**We plot the RCP8.5 results as suggested. There are some differences between two models,
so we agree with the reviewer that model difference is partially contaminating the
normalized results. We have fully acknowledged this caveat in Section 5 and also in the
conclusion section and called for future research to systemperically address this issue
(hopefully using multiple models).**

85-87 – missing citation to earlier work.

**Response:**
**Thanks, we added Lin et al., (2016a).**

90-91 – missing again.

**Response:**
**Thanks, we added Lin et al., (2016a).**

95-97 – it could also be that tropospheric aerosols are concentrated over the land.

**Response:**
**This is a great point. We now added, "...the stratospheric aerosols can be less effective in
changing surface energy balance than tropospheric aerosols, which can induce warm cloud
changes and are more concentrated over the land."**

06-07 – rather than reversing, I'd recommend "changing the sign of" or some other construction that's less ambiguous.

**Response:**
**We change "reversing" to "flipping". We also checked other places for consistency. This is a great suggestion.**

Table 5 – The formatting is not great, I don't think the X / Y format is the best choice.

**Response:**
**We struggle to find a better alternative without making it too complex.**
**Also, the format was used in Table 3 as well.**

The simulated PET value should also be reported as well as a total or mismatch column.

**Response:**
**The total change in PET in (%/°C) is shown in figure 8 and quotes extensively in the text already**.

18-22 – This paragraph should be revisited.

**Response:**
**It's rewritten as:**
*To summarize, the weaker sensitivity of Sulfur Injection globally, and the weaker response in absolute values over the Amazon region as shown in Section 4, are the two major counterarguments against the effectiveness of Sulfur Injection. The comparison with our earlier studies on tropospheric SO2 and volcanic eruption using different model configurations (the 20th century or Last Millennium; CESM with 1° or 2° resolution) demonstrate qualitative robustness of the results and the common physical mechanisms.*

26 – this is not a counter-argument, P/PET is a common measure of aridity P is not.

**Response:**
**We remove the "counteragrument" statement.**

29-33 – The experiments are very different, this paragraph describes them as if they had the same goals.

**Response:**
**As in response to general comments, we do think they are comparable because they share the same goal of stabilization climate at a predetermined level.**

35 – I'd suggest "less effective at offsetting the amazon drying"?

**Response:**
**Great. We change reverting to offsetting.**

40-47 – This difference in climate sensitivity also undermines the normalization procedure in section 5.

**Response:**
**We do not dispute that the difference in climate sensitivity is still perplexing the normalized results, but the normalization procedure is specifically conducted to mitigate the caveat of model difference.**

---

## Author Comment (AC3) · 1 Jun 2020

In this study, the authors compared model-simulated hydrological cycle change in two scenarios: a scenario in which global mean temperature, equator-to-pole temperature gradient, and interhemispheric temperature gradient, are all stabilized at present day level under the RCP8.5 background scenario through stratospheric sulphate aerosol injection (GLENS ensemble simulations); and a scenario in which atmospheric $CO_2$ is reduced to achieve the temperature stabilization goal of 1.5 degree (carbon capture and storage). The stratospheric sulphate injection simulations are done with CESMWACCM, and the carbon capture and storage simulations are done using CESM1- CAM5. The main metrics used in the analysis are precipitation (P), potential evapotranspiration (PET), and the ratio of P to PET. The regions focused on is North and South America.

**Response:**
**Thanks for the excellent summary.**

My biggest concern is to what extend hydrological cycle change in these two scenarios can be compared with each other. By experiment design, compared to the present-day climate, the global mean temperature is near zero in the GELENS, and 1.5 degree warming in carbon capture and storage simulation. Different amount of temperature change would certainly be one of the major factors responsible for different hydrological cycle change.

**Response:**
**Very quick clarification. The 1.5C warming is relative to the pre-industrial era, so the stabilized warming is less than 0.5C compared with present-day in the CCS simulation. Also, our comparison is mainly based on the "avoided warming" relative to RCP8.5 at the end of the 21st century. We did not directly analyze the change between now and the end of the 21st century, because, as the reviewer correctly pointed out, the GLENS experiment is designed to minimize such a change.**

The authors also presented changes that are normalized by global mean temperature change, but to what extend these hydrological metrics, in particular PET, scales with global mean temperature at the regional scale?

**Response:**
**Yes. The normalization by global mean temperature change (Section 5) is conducted to mitigate the difference in "avoided warming". The scalability of PET to temperature appears to be strong, even at regional level. In Figure 9 (scatter plot), the correlation is strong with R>0.95.**

CESM-WACCM and CESM1-CAM5 also has different model configuration and climate sensitivity, which further complicates the comparison between two sets of simulations.

**Response:**

**This is a good point, also brought up by another reviewer. The different impacts of these two geoengineering schemes are what we set out to quantify. Therefore, we must address the limitation of the current experiment set up – two large ensembles are from two related but different climate models.**
**As we now increasingly emphasized in this revision, we highlight four approaches to minimize this limitation:**
**(a) bias correction (see more technical description in this revision),**
**(b) normalization (Section 5 as the reviewer acknowledged, also heavily revised),**
**(c) interpretation of physical mechanism by breaking down PET, P/PET changes to individual climatic drivers, to highlight the role of solar dimming at the ground surface, and the shift of tropical rainfall out of deep tropics (Amazon), both of which are only strongly operating in the sulfate injection case (Section 4)**
**(d) further corroboration of the physical mechanisms at play, using other previously published simulations including volcanic eruption and tropospheric aerosols (Section 5).**

In the conclusion, the author states that 'As a result, we emphasize that the main purpose of this paper is not to examine the effectiveness of these two climate engineering schemes in the sense of absolute values. Instead, we aim to highlight the physical mechanisms at play, especially when distinct between the two approaches'. But most of the study is actually devoted to the comparison of these two scenarios quantitatively, and I really don't see a clear presentation of the fundamental physical mechanisms gained from this study.

**Response:**
**Most of the quantitative comparison is done thru relative values (fractional changes avoided by these two types of geoengineering).**
**We have tried to improve the presentation of the physical mechanisms in this revision (related to bullet points (c) and (d) in the response above), including the revised Table 5, Figure 4, Figure 7, Figure 8 and Figure 9. Further suggestions are certainly welcome.**

Specific comments:

Abstract: 'these two leading geoengineering schemes have not been carefully examined under a consistent numerical modelling framework.' Does it imply that this study is the first to carefully examine these two schemes in a consistent modelling framework? This is clearly not true. 'Here we present a comprehensive analysis of climate impacts . . .' This is not true. This study only analyzes some hydrological metrics for some specific regions. This is not a comprehensive analysis.

**Response:**
**Some previous work has studied the difference between sulfate injection and carbon dioxide increase/decrease. While the carbon capture impact should be similar to deep emission cut, the current design puts that to extreme by introducing a large amount of negative emission towards the later half of the 21st century, much higher than assumed in RCP2.6 (Figure 4e, solid green vs. dash green) .**
**We have tuned down the language to be**
**"...these two leading geoengineering schemes have not been directly compared under a consistent analytical framework using global climate models."**
**"Here we present the explicit analysis of hydroclimate impacts of these two geoengineering approaches at global and regional level"**

Introduction:

Line 64: The reference of Xu and Ramanathan, 2017 and Miller et al., 2017 is missing in the reference list. (please also check other references. Quite a few are missing in the reference list)

**Response:**
**Thanks, and we apologize for those mistakes and will check the reference list thoroughly.**

Line 64-65: Climate engineering is proposed as a potential method to mitigate global warming, but it is a too strong statement saying that climate engineering is needed in climate mitigation. In fact, in the abstract of Lawrence et al. (2018), as cited here, it states: "Based on present knowledge, climate geoengineering techniques cannot be relied on to significantly contribute to meeting the Paris Agreement temperature goals"

**Response:**
**Yes. We have now clarified what we meant:**
**"aggressive climate engineering schemes are required in order to meet these low-warming targets".**
**This is in line with Laweren et al, which basically said climate geoengineering is not sufficient in itself.**

Line 65: Please provide reference for these approaches. In particular, what is 'spraying sea water over sea ice' and 'oceanic evaporation enhancement' approaches?

**Response:**
**Yes, these are two less commonly discussed proposals. Both are highly controversial and are included for completeness.**
**Field, L., Ivanova, D., Bhattacharyya, S., Mlaker, V., Sholtz, A., Decca, R., et al. (2018). Increasing Arctic sea ice albedo using localized reversible geoengineering. Earth's Future, 6, 882–901. https:// doi.org/10.1029/2018EF000820**
**Salter, Stephen. "Spray turbines to increase rain by enhanced evaporation from the sea." Tenth Congress of International Maritime Association of the Mediterranean, Crete. 2002**

Line 67-70: It is confusing to state that they are global-scale schemes. In theory, each of the schemes described here can be implemented at either global or local scales.

**Response:**
**Right. In terms of impact, these two approaches, even when implemented at local scale, are less confined to regional levels, such as cloud brightening or land albedo modification. We change it to "two schemes that can have global impact …"**

Line 70: This sentence needs some rewriting. Stratospheric sulphate injection is usually considered to be relatively inexpensive.

**Response:**
**We change it to "Both approaches, especially the first one (carbon capture), can be  expensive... "**

Method: In addition to fundamental difference between GLENS and CO2 mitigation, these two sets of simulations use different versions of CESM, which adds another uncertainty to the results presented here.

**Response:**
**The difference of the two mitigation is what this analysis is aiming for, so we do not see that as a weakness. But in this revision, we have fully acknowledged the two similar but different versions of GCMs is a limitation of this pilot work. We have tried to mitigate this caveat (see responses previously to the general comments) and also call for future analysis using more models.**

By just reading the paragraph of the carbon capture and storage experiment, it's not clear to me whether this is emission-driven or concentration driven. It says net emission is reached at year 2050, and then says the corresponding CO2 concentration is prescribed rather than simulated.

**Response:**
**Sorry for the confusion. The CESM itself is concentration driven. The concentration is simulated using a simpler climate model (MiCES), prior to CESM simulation, which is emission driven. This is now clarified.**

Also, a figure showing the emission (or concentration) pathway for the carbon capture and storage experiment should be presented.

**Response:**
**We had shown the emission pathway (emission reduced) in Figure 4e.**
**We also add a comparison with RCP2.6 (which is well studied and compared with solar geoengineering) to show that the CCS scenario here is much more aggressive and contains a significant amount of carbon capture.**

2.4 Hydroclimate variables examined: The authors state: "we focus on climate quantities over land due to their close relevance to agriculture, ecosystems, and the carbon cycle ..". Why not also analyse some variables directly related to agriculture and carbon cycle, such as terrestrial gross and net primary production? They are available from CLM output.

**Response:**
**Good point, but we have tried to limit the metric in discussion to P/PET by comparing it with a few other drought indicators. This is to avoid too much complexity in the presentation.**
**We agree that CLM land output should be further analyzed to make a stronger link with the agriculture and carbon cycle. We considered it carefully but decided not to pursue that route at this stage, considering since the CLM version is also slightly different between the two models which will bring further complexity.**

Page 6, line 71: "Climate model output cannot be taken at its face value". This statement is not true. It depends on purpose. For climate modelling studies that aim to understand fundamental mechanisms, no bias correction is needed at all.

**Response:**
**True. We adjust it to be "Climate model output cannot be taken at its face value, especially for future projection."**
**Another benefit of bias correction is that it mitigates the model differences.**

3. Mitigation at the global scale
Most of this section is devoted to the presentation of numerical values and the characteristics for temporal evolutions of temperature, precipitation, and potential evapotranspiration. This kind of discussion should be shortened and replaced by scientific discussions of the underlying mechanisms.

**Response:**
**We will shorten Section 3 as suggested.**

It is not surprising that temperature response to SAI is quicker than that to carbon capture because of the long timescale associated with CO2 forcing, but I think more analyse should be done for the change in hydrological cycle.

**Response:**
**Thanks for recognizing a key message of Section 3. The long time scale of CO2 forcing is now presented in an improved version of Fig 4 (bottom row).**

How does the change in PET compare the change in soil moisture as presented by Cheng et al. (2019)?

**Response:**
**In Figures 1 and 2, we had compared P/PET change with soil moisture as studied in detail by Cheng et al., (2019). They are generally consistent but with notable regional differences.**

For carbon capture simulation, what is the role of the direct CO2 effect on land through the influence on stomatal opening, leaf area index, and vegetation dynamics (if any)?

**Response:**
**The Ball-Berry stomatal conductance model was utilized in both CLM4.0 and CLM4.5. though there is an improvement of the iterative solution in the stomatal conductance model under CLM4.5 relative to CLM4. The stomatal conductance is directly influenced by the relative humidity, the $CO_2$ concentration at the leaf surface, and the soil water.**
**The competition of all these terms are all systemically studied in Cheng et al., (2019), which found that under a warming climate, increases in all column soil liquid water and canopy intercepted water overwhelm the direct physiological effect of $CO_2$ (which tends to suppress the transpiration), and thus lead to an overall increase of canopy transpiration under RCP8.5 (Red line in the next figure).**

[Figure]

Anyway, just to present numbers does not help much to improve our scientific understanding.

**Response:**
**Again, we very much appreciate the suggestions of digging into land component model output and carbon cycle response. But that's really beyond the scope of this analysis which focuses on hydroclimate metrics. We do have a project in planning to look at land response more closely in agriculture regions with a focus on crop yields under various mitigation scenarios.**
**As for the scientific mechanisms, we now enhance the discussion of P/PET changes due to five governing drivers in Section 5 (Figure 10) and also contrasting that to previous model analogs of tropospheric GHGs, aerosols, and volcanic eruption. To our knowledge, this type of synthesis is not done before.**

3.1 temperature The lengthy discussion of global temperature change does not really provide any scientific insight. All it says it that SAI stabilize temperature change and carbon capture maintains 1.5 degree warming by the end of this century, both of which are achieved by experiment design. We can just use a few sentences to cover this info.

**Response:**
**We have shortened the temperature results in Section 3.1 to be two paragraphs.**

Page 9 line 43: What is 'carefully introduced'?

**Response:**
**We clarified it to be :**
**when the amount of Sulfur Injection is carefully adjusted to balance the CO2 warming**

Page 9, line 47: "Because of the careful experiment design here'. Does it imply that previous experiments are not carefully designed? Please rephrase.

**Response:**
**Change it to "In the current experiment design.."**

Page, 10, lines 87-88 "For example, the mid-century projected PET increase is 102.2 mm/year, but the Sulfur Injection can lower that by 127.8 mm/year, which drops the absolute value of PET by 25.6 mm/year" 102.2 mm/year is lowed by 127.8 mm/year? Please check the math and expression here.

**Response:**
**The math is correct and actually a point we wanted to make. Sulfur Injection can flip the sign of PET change from increase to decrease.**
**Clarified as "the mid-century projected PET increase is 102.2 mm/year, but the Sulfur Injection can reduce that by 127.8 mm/year, which actually leads to a drop of the absolute value of PET by 25.6 mm/year."**

Regional change: How does the presented PET change for GLENS compare with soil moisture change presented by Cheng et al. (2019)? This should be discussed.

**Response:**
**PET itself depicts the evaporation demand of the atmosphere and has no direct bearing on soil moisture. P/PET is more related to soil moisture and vegetation growth as shown by many studies previously. Thus, we only showed P/PET compared with soil moisture in Fig 1 and 2.**
**PET change is also similar to the ET changes as studied in Cheng (2019), not surprisingly because they are based on the same dataset.**

[Figure]

Page 15, lines 50-53: As clearly stated here, there is no direct comparison between GLENS and carbon capture experiment here. First of all, temperature change is not the same, which masks the usefulness of this study.

**Response:**
**True. If the experiment were perfectly designed, the temperature change should be maintained the same for both cases to facilitate a fair comparison. We acknowledge the caveat and call for more work into it.**
**However, retrospectively, we do not really see the temperature level difference of 0.5-1C as a major limiting factor. Several other comparison studies based on climate models are not producing perfecting aligned temperature trajectory either, for example Muri et al., (2018) with Figure 2b copied below. Also see Figure 3 of Niemeier et al., (2013).**

[Figure]

**Niemeier, U., Schmidt, H., Alterskjær, K., and Kristjánsson, J. E. ( 2013), Solar irradiance reduction via climate engineering: Impact of different techniques on the energy balance and the hydrological cycle, J. Geophys. Res. Atmos., 118, 11,905– 11,917**

**Muri, H., J. Tjiputra, O.H. Otterå, M. Adakudlu, S.K. Lauvset, A. Grini, M. Schulz, U. Niemeier, and J.E. Kristjánsson, 2018: Climate Response to Aerosol Geoengineering: A Multimethod Comparison. J. Climate, 31, 6319–6340, https://doi.org/10.1175/JCLI-D-17-0620.1**

5. Normalized change To what extend PET change scales with global mean temperature change at regional scale?

**Response:**
**See the response to a general comment.**

Page 16, lines 71-75: There are many studies on the different precipitation sensitivity to CO2 and aerosol forcing. The authors should discuss some of them, in addition to their own study (Lin et al., 2016)

**Response:**
**Right. We included a few more studies on precipitation response to both sulfur injection (Muri et al., 2019; Niemeier et al., 2013; Cao et al., 2017)  and to troposphere SO2 forcing (Ming et al., 2007)**

Page 16, lines 76-77: I don't understand how this conclusion is drawn. In this paragraph, only CESM model is mentioned.

**Response:**
**The earlier argument is flawed and we have removed the sentence in question.**

Page 16, lines 83-84: I just don't understand this sentence.

**Response:**
**Thanks for catching that. The related sentences are revised to be "Different from the precipitation sensitivity, the PET sensitivity to Carbon Capture and Sulfur Injection are similar (3.8 %/ºC vs. 4 %/ºC). A similar PET change, when combined with the greater precipitation decrease, will lead to a smaller increase in P/PET in response to Sulfur Injection. In other words, the almost identical PET sensitivity in response to Sulfur Injection and Carbon Capture is the main reason that Sulfur Injection has a smaller control over P/PET by a factor of two (Figure 9, -1.1 %/ºC vs. -2.2 %/ºC)."**

Page 17, lines 9-10: Does the contribution from different climate variables to PET add linearly to their combined effect?

**Response:**
**Yes. Because the composition is done with partial differentiation, they add up linearly to the total PET change.**

Page 17, line 18: What does the weaker sensitivity of sulphur injection mean? It should be 'the sensitivity of XX to sulphur injection is weaker'.

**Response:**
**Changed to "weaker sensitivity of P/PET in response to Sulfur Injection".**

Page 19, lines 56-60: "Instead, we aim to highlight the physical mechanisms at play, especially when distinct between the two approaches (e.g., radiative balances and dynamic response)." I really don't see what insightful physical mechanisms are highlighted in this study.

**Response:**
**We add a summary sentence here, supported by more revisions in the text. "The notable distinction between the two approaches include response time scales, the role of solar dimming at the surface, the shift of deep tropical rainfall. The direct physiological role of CO2 is potentially important because the CO2 level is greatly reduced in only one of the two mitigation approaches. But this study, focusing on meteorological drivers of land aridity using P/PET, did not delve into CO2's suppression on plant transpiration via stomatal closure, which also appears to be weak in these two models compared with other climate models (Swann et al., 2018)."**

---

## Author Response (AR1)

Editor Comments:
The following comments are on the original submission.

**Response:**
**Thanks very much for the comments. We have made further revisions and is now submitting the revision.**

1. I see that most of the comments of the reviewers are addressed. However, responses are given only to the reviewer for some of the comments and not addressed in the manuscript. I suggest the authors to address every comment in the manuscript.

**Response:**
**In this revision, all points raised by the three reviewers are addressed with point-to-point response. For some comments, responses are given on why no changes are made to the manuscript.**

2. Table 2, caption: Discuss the numbers in the parenthesis.

**Response:**
**We added to the caption: "Values in the parentheses are percentage relative to present-day levels."**

3. Tables 2, 4 and 5: Uncertainty estimates are missing in the current tables. It is important to provide them.

**Response:**
**For Table 2 and 4, we now add the uncertainty estimate based on the large ensemble spread (one standard deviation) after the +/- sign.**
**For Table 5, the presentation becomes too crowded with the uncertainty range, so we did not add it.**

4. Fig. 2 caption: suggest to change "dashed regions" to "stippling". Also, indicate the significance level in the caption, e.g. 5% or 10% significance level.

**Response:**
**Changed to "The areas with stippling in the panels of "End-of-Century – Present-day" are where the differences are significant at a 95% confidence level following the student's T-test."**

5. Figure 9: The top 3 panels are missing in the submitted file.

**Response:**
**In this submission, the top 3 panels for RH2M are included. This is now Fig 10.**

6. Figure 9, Caption: "Row a (left) is the changes in the baseline warming. Row (b) is the mitigated change due to carbon capture. Row (c) is the mitigated change due to sulfur injection" should be changed to "Left panels show the changes in baseline warming simulations, middle panels show the changes due to carbon capture and right panels show the changes due to sulfur injection"

**Response:**
**Thanks. Changed to "Left panels show the changes in baseline warming. Middle panels show mitigated change due to carbon capture. Right panels show the mitigated change due to sulfur injection."**

7. Lines 540-560, the differing slopes for P change: This is related to the differences in the fast adjustments to the CO2 and stratospheric SO4 aerosol (or solar) forcings. If the P changes related to fast adjustments are removed, then the slope in the two cases should be similar.

**Response:**
**Point well taken.**

**The distinct features of fast response to GHGs (CO2) vs. aerosols (tropospheric and stratospheric), as nicely illustrated in previous studies including the Editor's, in a main motivation for us to probe into these two newly available climate mitigation large ensemble simulations. Also, here, we want to move further to discuss the potential distinction in PET due to different forcings.**

**We agree that the SST-related P changes (slow response) would be more forcing independent. At this moment, we do not have plan to carry out additional simulation with fixed SST to tease out the fast and slow component. We had done similar analysis previously with fixed SST and SST-driven simulation to look at different tropospheric forcing terms.**

**Wang, Z., Lin, L., Yang, M., Xu, Y., and J. Li, (2017) Disentangling fast and slow responses of the East Asian summer monsoon to reflecting and absorbing aerosol forcings, Atmos. Chem. Phys., 17, 11075-11088, https://doi.org/10.5194/acp-17-11075-2017.**

A rich literature exists on this topic. You can discuss the role of fast adjustments in these differences and cite the following papers. I do understand that you would need fixed-SST experiments to diagnose these fast adjustment P changes.

G. Bala, P. B. Duffy, and K. E. Taylor, 2008: Impact of geoengineering schemes on the global hydrological cycle, Proceeding of the National Academy of Sciences, 105(22), 7664-7669

G. Bala, K. Calderia, R. Nemani, 2009: Fast versus slow response in climate change: Implication to the global hydrological cycle, Climate Dynamics, DOI 10.1007/s00382-009-0583-y

Duan, Long Cao, G. Bala and K. Caldeira, 2018: Comparison of the Fast and Slow Climate Response to Three Radiation Management Geoengineering Schemes, J. Geophys. Res. -Atmos., 10.1029/2018JD029034

**Response:**
**Thanks for suggesting these valuable refrences. We have included all of these in the text. Specifically, we add a discussion related to Fig 9:**

[revised manuscript text omitted]

**Page 12: [1] Deleted**                    **Xu, Yangyang**                    **6/8/20 10:38:00 PM**

**Page 12: [1] Deleted**                    **Xu, Yangyang**                    **6/8/20 10:38:00 PM**

**Page 12: [1] Deleted**                    **Xu, Yangyang**                    **6/8/20 10:38:00 PM**

**Page 12: [2] Deleted**                    **Xu, Yangyang**                    **6/8/20 10:38:00 PM**

**Page 12: [2] Deleted**                    **Xu, Yangyang**                    **6/8/20 10:38:00 PM**

**Page 12: [2] Deleted**                    **Xu, Yangyang**                    **6/8/20 10:38:00 PM**

**Page 12: [2] Deleted**                    **Xu, Yangyang**                    **6/8/20 10:38:00 PM**

**Page 12: [2] Deleted**                    **Xu, Yangyang**                    **6/8/20 10:38:00 PM**

**Page 12: [2] Deleted**                    **Xu, Yangyang**                    **6/8/20 10:38:00 PM**

**Page 12: [2] Deleted**                    **Xu, Yangyang**                    **6/8/20 10:38:00 PM**

**Page 12: [2] Deleted**                    **Xu, Yangyang**                    **6/8/20 10:38:00 PM**

**Page 12: [2] Deleted**                    **Xu, Yangyang**                    **6/8/20 10:38:00 PM**

**Page 12: [2] Deleted**                    **Xu, Yangyang**                    **6/8/20 10:38:00 PM**

**Page 12: [2] Deleted**                    **Xu, Yangyang**                    **6/8/20 10:38:00 PM**

**Page 12: [2] Deleted**                    **Xu, Yangyang**                    **6/8/20 10:38:00 PM**

**Page 12: [2] Deleted**                    **Xu, Yangyang**                    **6/8/20 10:38:00 PM**

| Page 12: [2] Deleted | Xu, Yangyang | 6/8/20 10:38:00 PM |
|---|---|---|

| Page 12: [2] Deleted | Xu, Yangyang | 6/8/20 10:38:00 PM |
|---|---|---|

| Page 12: [2] Deleted | Xu, Yangyang | 6/8/20 10:38:00 PM |
|---|---|---|

| Page 12: [2] Deleted | Xu, Yangyang | 6/8/20 10:38:00 PM |
|---|---|---|

| Page 12: [2] Deleted | Xu, Yangyang | 6/8/20 10:38:00 PM |
|---|---|---|

| Page 12: [2] Deleted | Xu, Yangyang | 6/8/20 10:38:00 PM |
|---|---|---|

| Page 12: [2] Deleted | Xu, Yangyang | 6/8/20 10:38:00 PM |
|---|---|---|

| Page 12: [2] Deleted | Xu, Yangyang | 6/8/20 10:38:00 PM |
|---|---|---|

| Page 12: [2] Deleted | Xu, Yangyang | 6/8/20 10:38:00 PM |
|---|---|---|

| Page 12: [2] Deleted | Xu, Yangyang | 6/8/20 10:38:00 PM |
|---|---|---|

| Page 12: [2] Deleted | Xu, Yangyang | 6/8/20 10:38:00 PM |
|---|---|---|

| Page 12: [2] Deleted | Xu, Yangyang | 6/8/20 10:38:00 PM |
|---|---|---|

| Page 12: [3] Deleted | Xu, Yangyang | 6/8/20 10:38:00 PM |
|---|---|---|

| Page 12: [3] Deleted | Xu, Yangyang | 6/8/20 10:38:00 PM |
|---|---|---|

**Page 12: [3] Deleted**            **Xu, Yangyang**            **6/8/20 10:38:00 PM**

**Page 31: [4] Deleted**            **Xu, Yangyang**            **6/8/20 10:38:00 PM**